# Tailoring the Training: Difficulty-Aware Learning Strategy Allocation for Large Language Models

Xiaoling Zhou [1 2]  Shuaiyu Zhou [1]  Zhemg Lee [3]  Tao Chen [2]  Xirui Li [2]  Peng Chen [2]
Jie Jiang [2]  Wei Ye [1]  Shikun Zhang [1]

## Abstract

Although reinforcement learning (RL) enhances the reasoning capabilities of large language models (LLMs), it is primarily learned from the model's self-generated distribution, limiting its ability to acquire reasoning skills beyond its initial knowledge. To overcome this, we propose a **D**ifficulty-**A**ware **L**earning **S**trategy **A**llocation (**DALSA**) framework, which adaptively assigns appropriate learning strategies to samples based on their difficulty signals. DALSA is built on the key insight that samples beyond models' knowledge scope are better addressed through supervised fine-tuning (SFT), while those within the boundary but insufficiently mastered benefit more from RL, and well-learned samples are discarded to avoid redundant updates. To realize this principle, we extract a series of difficulty-aware training characteristics and employ a learnable strategy allocator to dynamically determine the optimal learning strategy for each sample based on its training dynamics. The allocator and the LLM are alternately optimized, enabling adaptive strategy allocation. Furthermore, two regularization techniques, anti-curriculum weighting and adversarial label smoothing, are integrated to alleviate the inherent limitations of RL and SFT, backed by comprehensive theoretical analyses. Extensive experiments on ten LLMs ranging from 1.5B to 70B across various tasks indicate that DALSA consistently outperforms baselines under both full and parameter-efficient fine-tuning settings.

## 1. Introduction

Reinforcement learning (RL) has emerged as an effective paradigm for enhancing the reasoning capabilities of large language models (LLMs) (Zhou et al., 2025b; Jiang & Ferraro, 2026; Wang et al., 2026; Ma et al., 2026; Xu et al., 2025a). While achieving notable empirical performance, conventional RL typically relies on self-generated responses produced by the model through iterative rollouts and feedback (Shao et al., 2024). As a result, RL tends to refine and reinforce existing reasoning patterns supported by the model, optimizing behavior within the model's inherent knowledge space rather than explicitly introducing new factual information or expanding the underlying knowledge scope (Yue et al., 2025; Zhao et al., 2025).

By contrast, supervised fine-tuning (SFT) enhances LLM capabilities through explicit supervision (Li et al., 2025c; Dong et al., 2024). By using high-quality labeled data, SFT enables models to incorporate external knowledge and skills (Mecklenburg et al., 2024; Yue et al., 2025). Empirical evidence suggests that SFT can yield strong performance, particularly under limited model capacity or data-constrained settings (Guo et al., 2025). However, the effectiveness of SFT is limited by its reliance on large-scale, high-quality demonstrations, its susceptibility to overfitting, and its poor generalization to out-of-distribution (OOD) inputs, which reduces its robustness in downstream applications (Mecklenburg et al., 2024; Jiang et al., 2025b).

Accordingly, a growing body of research has explored the integration of SFT and RL to leverage their complementary strengths (Qiu et al., 2025; Yan et al., 2025; Ma et al., 2025). Despite their effectiveness, existing approaches are hindered by several notable limitations. First, most methods rely on fixed or manually determined allocations between SFT and RL, limiting their ability to dynamically adjust strategies in response to the evolving capabilities of LLMs throughout training (Chen et al., 2025). Second, the difficulty modeling is often coarse or incomplete, typically relying on a single type of indicator, such as entropy (Fu et al., 2025) or accuracy (Ma et al., 2025). This oversimplification fails to capture the multifaceted nature of learning difficulty, thereby constraining the efficacy of tailored op-

---

[1]Peking University, Beijing, China [2]Tencent, Beijing, China [3]Tianjin University, Tianjin, China. Correspondence to: Wei Ye <wye@pku.edu.cn>, Shikun Zhang <zhangsk@pku.edu.cn>.

*Proceedings of the 43rd International Conference on Machine Learning*, Seoul, South Korea. PMLR 306, 2026. Copyright 2026 by the author(s).

timization. Third, these approaches generally overlook the intrinsic limitations of SFT and RL, such as overfitting and a bias toward easier samples, further restricting their potential for performance improvement (Wang et al., 2024b).

To address these challenges, we propose a novel **D**ifficulty-**A**ware **L**earning **S**trategy **A**llocation (**DALSA**) framework that dynamically assigns learning strategies, including SFT, RL, or DISCARD, to individual samples based on their difficulty signals. The core principle of DALSA is that

"*Samples that lie outside the model's knowledge scope are more effectively learned via SFT to facilitate new knowledge acquisition, samples that fall within the model's knowledge boundary but remain insufficiently optimized are better refined through RL, and samples fully mastered can be discarded to avoid redundant optimization.*"

Specifically, DALSA extracts a series of training characteristics, such as perplexity and entropy, from the LLM to capture sample difficulty. These features are subsequently fed into a strategy allocator, which produces a probability distribution over candidate learning strategies for each sample. The strategy with the highest predicted suitability is then applied to guide the learning of the corresponding sample, enabling a fine-grained and adaptive optimization process. Furthermore, an anti-curriculum weighting scheme for RL and an adversarial label smoothing (ALS) loss for SFT are incorporated to mitigate their inherent limitations, with theoretical analysis supporting their effectiveness in improving model convergence and robustness. The LLM and the allocator are optimized alternatively, allowing strategy assignment to evolve dynamically throughout training. Extensive experiments across various LLMs and tasks, under full and parameter-efficient fine-tuning (PEFT) settings, indicate that DALSA consistently outperforms existing baselines in enhancing the reasoning capability of LLMs.

Overall, our main contributions are as follows:

- We introduce DALSA, a novel training framework for LLMs that adaptively allocates learning strategies to individual samples based on their difficulty signals.

- We propose complementary regularization techniques for RL and SFT, namely, anti-curriculum weighting and ALS loss, and provide theoretical analyses demonstrating their effectiveness.

- We conduct extensive experiments across various LLMs and tasks, assessing full fine-tuning and PEFT settings, which highlight DALSA's *state-of-the-art* performance in enhancing LLMs' reasoning capabilities.

## 2. Preliminary

**Learning Difficulty of Samples**   Learning difficulty is an intrinsic property of samples (Zhou & Wu, 2023; Xu et al.,

2025b). For LLMs, it is typically characterized by several types of metrics (Zhou et al., 2025a; 2023; Li et al., 2025a; Zhang et al., 2025): loss-based (Gao et al., 2025; Chen et al., 2024b; Li et al., 2023), perplexity-based (Ankner et al., 2024; Li et al., 2024b), reference model-based (Liu et al., 2025a; Ye et al., 2025), and length-based measures (Muennighoff et al., 2025). Loss-based metrics typically estimate the learning difficulty of samples using training or validation loss (Gao et al., 2025; Chen et al., 2024b; Yang et al., 2026b), while perplexity has been widely adopted in applications such as pretraining data selection and instruction tuning (Ankner et al., 2024; Li et al., 2024b). Reference model-based approaches assess sample difficulty by comparing model behaviors across one or multiple LLMs, for example, through accuracy distributions (Liu et al., 2025a; Ye et al., 2025). Additional indicators include the number of optimization steps required for model alignment (Gao et al., 2025) and the length of reasoning traces in mathematical problem-solving tasks (Muennighoff et al., 2025). Despite their effectiveness, existing difficulty metrics primarily rely on isolated indicators, limiting their ability to capture the full training dynamics of samples and hindering their effectiveness in tailored optimization during LLM training.

**Integration of RL and SFT**   Recent research has investigated the integration of SFT and RL to harness their complementary strengths (Guo et al., 2025; Wen et al., 2025). For instance, UFT (Wang et al., 2024b) unifies SFT with RL-based methods like RLHF (Christiano et al., 2017) and DPO (Rafailov et al., 2023), addressing catastrophic forgetting via implicit reward formulation. Metis-RISE (Qiu et al., 2025) activates latent reasoning with RL, followed by SFT to refine under-learned reasoning paths. LUFFY (Yan et al., 2025) improves RL with off-policy reasoning traces, balancing imitation learning and exploration through the incorporation of off-policy demonstrations. ReLIFT (Ma et al., 2025) proposes an interleaved fine-tuning framework where RL is the primary optimization method, and challenging samples are handled through SFT. Similarly, TAPO (Wu et al., 2025) enhances RL with external "thought patterns" to balance exploration and strategy exploitation. SASR (Chen et al., 2025) adapts SFT or RL usage based on model gradients, while SRFT (Fu et al., 2025) combines both through entropy-aware weighting, facilitating global policy refinement. Despite their empirical advantages, existing methods often rely on predefined schedules or heuristic allocation strategies, as well as lack a comprehensive modeling of sample difficulty, which limits their adaptability and efficacy in learning heterogeneous samples.

## 3. Methodology

This study presents DALSA, a novel training framework for LLMs, which employs a learning strategy allocator

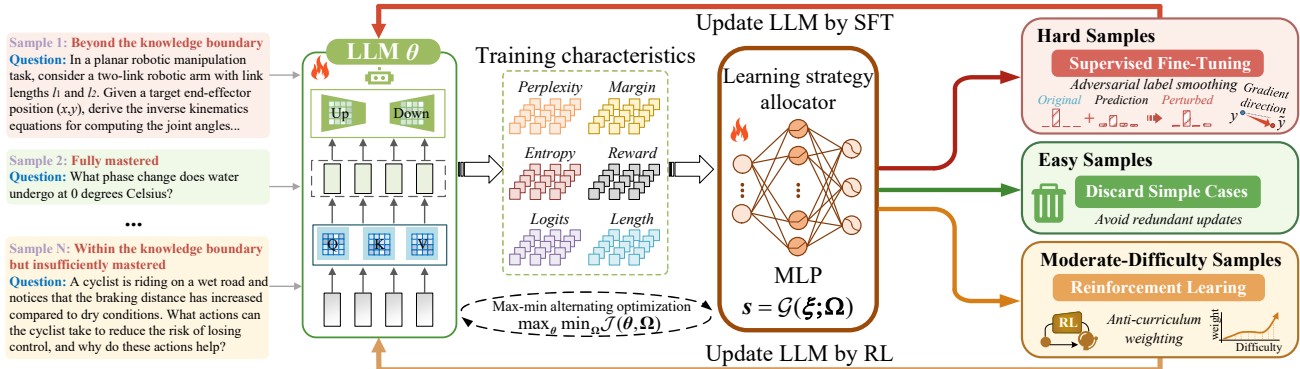

*Figure 1.* The diagram of the DALSA framework. Training characteristics are first extracted from the LLM and fed into the learning strategy allocator to determine the appropriate learning strategy for each sample: SFT, RL, or DISCARD. To enhance SFT, an ALS loss is employed to mitigate overfitting, while RL integrates a dynamic anti-curriculum mechanism to reduce bias toward easier samples. The allocator is optimized jointly with the LLM to ensure that the allocation process aligns with the LLM's evolving training dynamics.

to dynamically assign sample-specific strategies, such as **SFT**, **RL**, or **DISCARD**, based on their distinct training characteristics. Additionally, the framework integrates an anti-curriculum weighting scheme and an ALS loss to mitigate the inherent limitations of RL and SFT training. An overview of our entire framework is illustrated in Fig. 1.

### 3.1. Unified Optimization Objective

SFT updates model parameters by maximizing the log-likelihood of fixed training examples, which can be regarded as a special case of off-policy RL, where the reward is constant, and learning is confined to predefined trajectories. In contrast, RL optimizes the expected reward of the policy through on-policy sampling and policy gradient methods, facilitating adaptive exploration within the model's output space. To capitalize on the complementary strengths of SFT and RL, we propose a unified optimization objective for our training framework, incorporating two sample-specific binary coefficients to enable flexible and adaptive integration.

$$\mathcal{J}\left(\boldsymbol{\theta}, \boldsymbol{\Omega}\right) = \mathbb{E}_{\boldsymbol{x} \sim \mathcal{D}}\left[\alpha_{\boldsymbol{x}}\left(\boldsymbol{\Omega}\right) \mathcal{J}_{\boldsymbol{x}}^{\text{SFT}}\left(\boldsymbol{\theta}\right) + \beta_{\boldsymbol{x}}\left(\boldsymbol{\Omega}\right) \mathcal{J}_{\boldsymbol{x}}^{\text{RL}}\left(\boldsymbol{\theta}\right)\right],$$
$$(1)$$

where the SFT and RL objectives are defined as follows:

$$\mathcal{J}_{\boldsymbol{x}}^{\text{SFT}}\left(\boldsymbol{\theta}\right) = \log \pi_{\boldsymbol{\theta}}\left(\boldsymbol{y}^{\text{ref}} \mid \boldsymbol{x}\right), \quad (2)$$

$$\mathcal{J}_{\boldsymbol{x}}^{\text{RL}}(\boldsymbol{\theta}) = \mathbb{E}_{\boldsymbol{y} \sim \pi_{\boldsymbol{\theta}}(\cdot|\boldsymbol{x})}\left[\mathcal{A}(\boldsymbol{x}, \boldsymbol{y})\right]. \quad (3)$$

Here, $\pi_{\boldsymbol{\theta}}$ denotes the LLM policy, $\mathcal{A}(\boldsymbol{x}, \boldsymbol{y})$ represents the advantage function, and $\boldsymbol{y}^{\text{ref}}$ refers to the reference answer[1]. The two binary coefficients, $\alpha_{\boldsymbol{x}}$ and $\beta_{\boldsymbol{x}}$, dictate the learning strategy for each sample. Specifically, a sample is optimized using SFT when $\alpha_{\boldsymbol{x}} = 1$ and $\beta_{\boldsymbol{x}} = 0$, optimized using RL when $\alpha_{\boldsymbol{x}} = 0$ and $\beta_{\boldsymbol{x}} = 1$, and discarded when both $\alpha_{\boldsymbol{x}}$ and $\beta_{\boldsymbol{x}}$ are zero. These coefficients are not manually set but

---

[1]The reference answer $\boldsymbol{y}^{\text{ref}}$ can be obtained either from a more capable reasoning model or from human experts (Ma et al., 2025).

are adaptively determined by a learnable strategy allocator parameterized by $\boldsymbol{\Omega}$, as detailed in the subsequent section.

### 3.2. Learning Strategy Allocation

Prior research has demonstrated that SFT is particularly effective for samples that lie beyond the model's current knowledge or reasoning capabilities (i.e., challenging samples), whereas RL is more suitable for refining responses within the model's existing competence range (Ma et al., 2025; Qiu et al., 2025). Therefore, *it is essential to assign distinct learning strategies to samples based on their inherent difficulty signals* (Zhou & Wu, 2023; Chen et al., 2024a). As discussed in Section 2, the difficulty of a sample can be quantified using various training dynamics, such as perplexity and prediction distributions. In this context, we extract a series of training characteristics for each sample from the LLM and automatically learn the mapping between these features and the corresponding optimal learning strategies. The specific characteristics extracted are detailed as follows:

(i) **Margin** quantifies the model's ability to discriminate among predictions. We compute both the Top-1 and Top-$K$ margins, with $K = 10$. The Top-1 margin is defined as $p^1(y_t|\boldsymbol{x}, \boldsymbol{y}_{<t}) - p^2(y_t|\boldsymbol{x}, \boldsymbol{y}_{<t})$, while the Top-$K$ margin is $\frac{2}{K(K-1)} \sum_{i=1}^{K-1} \sum_{j=i+1}^{K}[p^i(y_t|\boldsymbol{x}, \boldsymbol{y}_{<t}) - p^j(y_t|\boldsymbol{x}, \boldsymbol{y}_{<t})]$, where $K > 1$. Here, $\boldsymbol{p}\left(y_t \mid \boldsymbol{x}, \boldsymbol{y}_{<t}\right)$ represents the predicted probability distribution for the $t$-th position, conditioned on the input $\boldsymbol{x}$ and the preceding tokens $\boldsymbol{y}_{<t}$, and $p^i(y_t|\boldsymbol{x}, \boldsymbol{y}_{<t})$ denotes the $i$-th largest element in $\boldsymbol{p}(y_t|\boldsymbol{x}, \boldsymbol{y}_{<t})$.

(ii) **Perplexity** quantifies the model's predictive confidence, defined as $\sqrt[S]{\prod_{t=1}^{S} 1/p^1\left(y_t \mid \boldsymbol{x}, \boldsymbol{y}_{<t}\right)}$, where $S$ denotes the sequence length.

(iii) **Entropy** is a standard metric utilized to quantify the unpredictability of model predictions. It is computed

as $-\sum_k p(y_{t,k} \mid \boldsymbol{x}, \boldsymbol{y}_{<t}) \log p(y_{t,k} \mid \boldsymbol{x}, \boldsymbol{y}_{<t})$, where $p(y_{t,k} \mid \boldsymbol{x}, \boldsymbol{y}_{<t})$ represents the predicted probability for the $t$-th position of the $k$-th token.

(iv) **Reward mean** $\mathbb{E}_{\boldsymbol{y}}[\mathcal{R}(\boldsymbol{x}, \boldsymbol{y})]$ quantifies the average feedback performance. A higher reward mean indicates that the generated responses yield favorable outcomes, reflecting that the model has attained a high level of proficiency with respect to the current sample.

(v) **Logits** refer to the hidden states prior to the probabilities. The logits norm quantifies the alignment between the model's deep features and the classifier weights, serving as an indicator of prediction confidence.

(vi) **Sequence length** can also serve as an indicator of the learning difficulty, where longer sequences are frequently more complex and more challenging to learn.

In the above formulation, token-level indicators are aggregated by averaging across the entire sentence, yielding corresponding sample-level values. The concatenated characteristics, denoted as $\boldsymbol{\xi}$, are subsequently input into a learning strategy allocator $\mathcal{G}$, implemented as a two-layer multilayer perceptron (MLP), to determine the appropriate learning strategy for each sample. Specifically, the allocator outputs a three-dimensional vector $\boldsymbol{S} = \mathcal{G}(\boldsymbol{\xi}; \boldsymbol{\Omega})$, where $\boldsymbol{\Omega}$ represents the MLP parameters, and $\boldsymbol{S} = \left[s^{\mathrm{SFT}}, s^{\mathrm{RL}}, s^{\mathrm{D}}\right]$ corresponds to the probabilities of selecting SFT, RL, or discarding the sample, respectively[2]. Then, the two binary coefficients can be specified as

$$\alpha = \mathbb{I}\{\arg \max_{k \in \{\mathrm{SFT,RL,D}\}} s^k = \mathrm{SFT}\}, \qquad (4)$$

$$\beta = \mathbb{I}\{\arg \max_{k \in \{\mathrm{SFT,RL,D}\}} s^k = \mathrm{RL}\}, \qquad (5)$$

where $\mathbb{I}(\cdot)$ represents an indicator function. The proposed method for allocating learning strategies to samples is fully automated and data-driven, relying solely on the policy learned by the strategy allocator, thereby eliminating the need for manual thresholds or heuristic rules. Furthermore, the learning difficulty of samples can be directly quantified by the SFT assignment probability $s^{\mathrm{SFT}} \in [0, 1]$, which is entirely determined by sample characteristics and the model state. A higher $s^{\mathrm{SFT}}$ value indicates a greater need for SFT-based learning, implying that the model has not yet sufficiently mastered the knowledge encoded in the sample.

## 3.3. Max-Min Alternating Optimization

The training objective of our framework is formulated as a max-min optimization problem, $\max_{\boldsymbol{\theta}} \min_{\boldsymbol{\Omega}} \mathcal{J}(\boldsymbol{\theta}, \boldsymbol{\Omega})$, as defined in Eq. (1), where $\boldsymbol{\theta}$ and $\boldsymbol{\Omega}$ are updated in an alternate manner. Specifically, for a given training batch $\mathcal{B}$, the parameters of the LLM, $\boldsymbol{\theta}$, are optimized to maximize model

performance as follows:

$$\boldsymbol{\theta}^{t+1} = \boldsymbol{\theta}^t + \eta_1 \nabla_{\boldsymbol{\theta}} \mathbb{E}_{\boldsymbol{x} \sim \mathcal{B}} \left[ \alpha_{\boldsymbol{x}} \mathcal{J}_{\boldsymbol{x}}^{\mathrm{SFT}}(\boldsymbol{\theta}) + \beta_{\boldsymbol{x}} \mathcal{J}_{\boldsymbol{x}}^{\mathrm{RL}}(\boldsymbol{\theta}) \right], \qquad (6)$$

where $\eta_1$ is used to scale the gradients associated with the SFT and RL objectives. Afterward, with the parameters of the LLM, $\boldsymbol{\theta}$, held fixed, the parameters of the learning strategy allocator, $\boldsymbol{\Omega}$, are updated as follows:

$$\boldsymbol{\Omega}^{t+1} = \boldsymbol{\Omega}^t - \eta_2 \nabla_{\boldsymbol{\Omega}} \mathbb{E}_{\boldsymbol{x} \sim \mathcal{B}} \left[ \alpha_{\boldsymbol{x}}(\boldsymbol{\Omega}) \mathcal{J}_{\boldsymbol{x}}^{\mathrm{SFT}} + \beta_{\boldsymbol{x}}(\boldsymbol{\Omega}) \mathcal{J}_{\boldsymbol{x}}^{\mathrm{RL}} \right], \qquad (7)$$

where $\eta_2$ is the learning rate of the allocator. By minimizing the objective with respect to $\boldsymbol{\Omega}$, the allocator is guided to assign higher probabilities to learning strategies that currently underperform yet possess greater potential for improvement. Owing to the non-differentiability of the $\arg \max$ operation in Eqs. (4) and (5), we adopt the Straight-Through Estimator (Bengio et al., 2013) to enable gradient-based optimization by approximating the gradients of discrete strategy selection with those of its continuous relaxation. Through this alternating optimization, the learned allocation polices co-evolve with the LLM throughout training.

Furthermore, targeted enhancements are introduced to address the inherent limitations of RL and SFT strategies, which are detailed as follows:

**Anti-Curriculum Weighting for RL** Curriculum learning (Bengio et al., 2009), inspired by human pedagogy, encourages mastering simpler patterns before progressing to more complex ones. Noting that RL tends to favor easier samples due to their stable gradients and low variance (Greenberg et al., 2023), we propose a dynamic anti-curriculum weighting strategy that progressively shifts RL's focus toward more challenging samples. Specifically, during optimization, samples with higher estimated difficulty scores are assigned greater weights, thereby facilitating more balanced and effective learning across the full spectrum of sample difficulties. Theoretical results, as formalized in Theorem 3.1[3], establish the favorable convergence properties of the proposed anti-curriculum RL mechanism.

**Theorem 3.1.** *The objective function $\mathcal{J}^{\mathrm{RL}} \in \mathcal{J}_n$ of the policy gradient algorithm is a finite-sum Lipschitz-smooth function with constant $L$. The gradient of $\mathcal{J}^{\mathrm{RL}}$ is uniformly bounded by $\sigma$, and the positive weights $w$ are constrained within the interval $[\underline{w}, \overline{w}]$. Let the learning rate be defined as $\eta_k = \eta = c/\sqrt{K}$, where $c = \sqrt{\frac{2(\mathcal{J}^{\mathrm{RL}}(\boldsymbol{\theta}^*) - \mathcal{J}^{\mathrm{RL}}(\boldsymbol{\theta}^0))}{L\sigma^2 \overline{w}^2}}$ and $K$ is the total number of iterations. Moreover, $\rho$ denotes the sample difficulty, and $\boldsymbol{\theta}^*$ is an optimal solution. Then, the iterates of the anti-curriculum RL algorithm satisfy*

$$\min_{0 \le k \le K} \mathbb{E}\left[ \left\| \nabla \mathcal{J}^{\mathrm{RL}}(\boldsymbol{\theta}^k) \right\|^2 \right] \le \sqrt{\frac{2\left(\mathcal{J}^{\mathrm{RL}}(\boldsymbol{\theta}^*) - \mathcal{J}^{\mathrm{RL}}(\boldsymbol{\theta}^0)\right) L\overline{w}^2}{K\underline{w}^2}} \sigma - \omega, \qquad (8)$$

---

[2]To encourage the model to make more discriminative decisions, we utilize a $\psi$-Softmax layer: $\mathrm{Softmax}((\boldsymbol{w}\boldsymbol{X} + b)/\psi)$.

[3]The complete proof is provided in Appendix B.

where $\omega = \min_k \frac{C_k}{\bar{w}} \cdot \text{Cov}\left(w_k, \rho_k\right)$, *and $C_k$ is a positive constant specific to $k$. The anti-curriculum weights conforms to $\text{Cov}\left(w_k, \rho_k\right) > 0$. Consequently, the weighted optimizer is stabilized and converges to a stationary point at a rate of $\mathcal{O}(1/\sqrt{K})$. Moreover, this bound is tighter than those achieved when the weights are independent of, or negatively correlated with, sample difficulty, indicating accelerated convergence in terms of iteration complexity.*

Accordingly, as stated in Section 3.2, by treating the probability assigned to SFT training, $\rho = s^{\text{SFT}}$, as a measure of its difficulty, we propose the following computationally efficient weighting function for hard-sample prioritization, which satisfies the conditions outlined in the proof:

$$w_{\boldsymbol{x}} = \frac{(1 - \lambda(t)) + \lambda(t)\rho_{\boldsymbol{x}}}{\mathbb{E}_{\hat{\boldsymbol{x}} \in \mathcal{B}}\left[(1 - \lambda(t)) + \lambda(t)\rho_{\hat{\boldsymbol{x}}}\right]}, \quad (9)$$

where $\lambda(t)$ controls the strength of the hard-sample prioritization, and $w_{\boldsymbol{x}}$ represents the weight of sample $\boldsymbol{x}$. Setting $\lambda(t) \equiv 0$ recovers standard RL, whereas $\lambda(t) \equiv 1$ corresponds to a fully difficulty-driven reweighting scheme. To ensure stability in the early stages and gradually shift focus to harder samples, $\lambda(t)$ increases with training iterations, defined as $\lambda(t) = \lambda_{\max} \cdot (1 - \exp\left(-k \cdot t/\mathcal{T}\right))$, where $t$ is the current iteration, and $\mathcal{T}$ is the total number of iterations. The parameter $\lambda_{\max}$ sets the upper bound of $\lambda$, and $k$ controls its growth rate, with $\lambda_{\max} = 0.8$ and $k = 0.5$ in practice[4].

**Adversarial Label Smoothing for SFT**  Label smoothing is a regularization technique that reduces overconfidence in model predictions, enhancing generalization and mitigating overfitting. It involves adding a small perturbation $\delta_j$ to the target labels $y_j$ (Szegedy et al., 2016) as follows:

$$\tilde{y}_j = (1 - \vartheta) \cdot y_j^{\text{ref}} + \vartheta\delta_j, \quad \vartheta \in [0, 1], \quad (10)$$

where $y_j^{\text{ref}}$ and $\tilde{y}_j$ represent the original and smoothed ground-truth values at the $j$-th position; $\delta_j$ denotes the applied perturbation, and $\vartheta$ is the smoothing factor that controls the perturbation magnitude.

To effectively mitigate overfitting and enhance robustness against label noise, we propose the ALS loss, in which perturbations are adaptively derived from the model's predictive distribution to maximize the standard cross-entropy (CE) loss, $\mathcal{L}_{\text{CE}}$. Specifically, the gradient of $\mathcal{L}_{\text{CE}}$ with respect to $\tilde{y}_j$ is given by $\frac{\partial \mathcal{L}_{\text{CE}}}{\partial \tilde{y}_j} = -\log p_j$, where $p_j$ denotes the predicted probability of token $j$. Based on this gradient, the adversarial perturbation is defined as $\delta_j \propto -\tau \log p_j$, with $\tau$ controlling the sharpness of the perturbation. An exponential reparameterization is then employed to ensure feasibility under the probability simplex constraints and maintain smooth optimization, yielding $\delta_j = \exp\left(-\tau \log p_j\right) = p_j^{-\tau}$.

Subsequently, the perturbations are normalized across all tokens to form a valid probability distribution, and the resulting ALS loss is formulated as:

$$\mathcal{L}_{\text{ALS}} = -\sum_{j=1}^{|\mathcal{V}|}\left[(1 - \vartheta)y_j^{\text{ref}} + \vartheta\frac{p_j^{-\tau}}{\sum_v p_v^{-\tau}}\right]\log p_j, \quad (11)$$

where $|\mathcal{V}|$ denotes the vocabulary size. Notably, setting $\tau = 0$ reduces the ALS loss to conventional label smoothing (Szegedy et al., 2016), which uniformly distributes probability mass across classes, whereas $\vartheta = 0$ recovers the standard CE loss. By incorporating these adaptive adversarial perturbations, the ALS loss mitigates overconfidence and alleviates overfitting in SFT. Further analyses of the effectiveness of ALS are provided in Appendix C.

# 4. Experiments

We assess the effectiveness of the DALSA framework under two LLM updating scenarios: *full fine-tuning* and *PEFT* (Rao et al., 2025b). For full fine-tuning, experiments are conducted on five competition-level mathematical reasoning benchmarks as well as one OOD benchmark, using LLMs including Qwen2.5-Math-7B (Yang et al., 2024b), Qwen2.5-Math-1.5B, Qwen2.5-7B (Yang et al., 2024a), Qwen3-8B (Yang et al., 2025), and LLaMA3-8B (Grattafiori et al., 2024). For PEFT, evaluations are conducted on eight commonsense reasoning tasks, two math reasoning tasks, and two code generation tasks, utilizing LLaMA2-7B (Touvron et al., 2023), LLaMA3-8B (Grattafiori et al., 2024), Mistral-7B (Jiang et al., 2023), and Gemma-7B (Mesnard et al., 2024), as well as two larger-scale LLMs, LLaMA3-70B and Mistral-8×7B.

## 4.1. Full Fine-Tuning

**Setups**  Following prior research (Ma et al., 2025; Yan et al., 2025), our training set comprises a curated subset of OpenR1-Math-220k[5] (Bakouch et al., 2025), with prompts sourced from NuminaMath 1.5 (Li et al., 2024a) and demonstrations generated by DeepSeek-R1 (Guo et al., 2025). We employ the LUFFY (Yan et al., 2025)-filtered dataset, resulting in 46k prompts paired with high-quality demonstrations. GRPO (Shao et al., 2024) is utilized as the RL algorithm. Consistent with prior research (Ma et al., 2025; Yan et al., 2025), KL divergence is omitted during training, and both length normalization and standard error normalization are excluded from the GRPO loss.

The remaining hyperparameters are configured as follows: an update batch size of 64, a learning rate of $1 \times 10^{-6}$, and a rollout generation temperature of 1.0. The reward

---

[4]Sensitivity studies are included in Appendix K.

[5]https://huggingface.co/datasets/open-r1/OpenR1-Math-220k

*Table 1.* Overall results on five mathematical reasoning benchmarks and one OOD benchmark, evaluated using Qwen2.5-Math-7B. **Bold** and underlined values indicate the best and second-best performance, respectively. $^\dagger$ and $^*$ indicate results reported in LUFFY (Yan et al., 2025) and ReLIFT (Ma et al., 2025), respectively.

| Method | AIME-24 | | AIME-25 | | AMC | | MATH-500 | | Olympiad | | MMLU-Pro | | Overall | |
|---|---|---|---|---|---|---|---|---|---|---|---|---|---|---|
| | *ACC* | *LEN* | *ACC* | *LEN* | *ACC* | *LEN* | *ACC* | *LEN* | *ACC* | *LEN* | *ACC* | *LEN* | *ACC* | *LEN* |
| *Qwen2.5-MATH-7B* | | | | | | | | | | | | | | |
| Qwen-Math* | 16.4 | 2125 | 6.9 | 2039 | 45.5 | 1493 | 65.1 | 1171 | 29.7 | 1594 | 24.9 | 830 | 31.4 | 1542 |
| Qwen-Math-Instruct* | 9.3 | 4065 | 8.2 | 3590 | 40.5 | 2905 | 78.2 | 1774 | 36.2 | 2888 | 34.0 | 4274 | 34.4 | 3249 |
| *Previous RL Methods* | | | | | | | | | | | | | | |
| SimpleRL-Zero$^\dagger$ | 27.0 | - | 6.8 | - | 54.9 | - | 76.0 | - | 34.7 | - | 34.5 | - | 39.0 | - |
| OpenReasoner-Zero$^\dagger$ | 16.5 | - | 15.0 | - | 52.1 | - | 82.4 | - | 47.1 | - | **58.7** | - | 45.3 | - |
| PRIME-Zero$^\dagger$ | 17.0 | - | 12.8 | - | 54.0 | - | 81.4 | - | 40.3 | - | 32.7 | - | 39.7 | - |
| Oat-Zero$^\dagger$ | **33.4** | - | 11.9 | - | 61.2 | - | 78.0 | - | 43.4 | - | 41.7 | - | 45.9 | - |
| *Replication of RL and SFT* | | | | | | | | | | | | | | |
| RL* | 21.1 | 3287 | 17.5 | 2904 | 62.1 | 2127 | 85.7 | 1277 | 48.6 | 2178 | 46.3 | 1276 | 46.9 | 2175 |
| SFT* | 26.9 | 7344 | 25.5 | 7059 | 59.8 | 5675 | 84.8 | 3503 | 52.6 | 5662 | 44.4 | 3956 | 49.0 | 5533 |
| *Methods Combining RL and SFT* | | | | | | | | | | | | | | |
| RL w/ SFT loss* | 26.9 | 7344 | 23.1 | 7059 | 59.8 | 5675 | 84.1 | 3503 | 53.6 | 5662 | 44.4 | 3956 | 48.6 | 5508 |
| LUFFY* | 27.3 | 5741 | 23.0 | 5267 | 63.5 | 3477 | 85.6 | 2361 | 53.8 | 3758 | 52.6 | 2241 | 50.9 | 3808 |
| ReLIFT* | 28.3 | 5062 | 22.9 | 4571 | 65.1 | 3540 | 87.9 | 2241 | 57.3 | 3352 | 53.9 | 2248 | 52.6 | 3502 |
| DALSA (Ours) | 29.4 | 4801 | **26.0** | 4265 | **66.6** | 3551 | **88.5** | 2093 | **59.8** | 3395 | 55.5 | 2079 | **54.3** | 3364 |

function follows the ReLIFT protocol, employing a binary scheme that assigns a reward of 1 for correct predictions and 0 for incorrect ones. For SFT, we use a batch size of 64, a learning rate of $5 \times 10^{-5}$, and train for three epochs. The hyperparameters introduced by DALSA are set to $\lambda_{\max} = 0.8$ and $k = 0.5$ in all experiments. For the ALS loss, the smoothing strength $\vartheta$ and perturbation sharpness $\tau$ are fixed at 0.005 and 0.5, respectively. The strategy allocator is implemented as a two-layer MLP with hidden dimensions of 256 and 128 units, and is trained using a learning rate of $1 \times 10^{-3}$. Further details are provided in Appendix E.

**Baselines** In line with Ma et al. (2025), we first compare the proposed DALSA framework with four representative RL approaches: SimpleRL-Zero (Zeng et al., 2025), which trains from Qwen2.5-Math-7B with rule-based reward signals; Oat-Zero (Liu et al., 2025b), which introduces modifications to GRPO by removing the standard deviation in advantage computation and applying token-level normalization in the policy loss; PRIME-Zero (Cui et al., 2025), which employs policy rollouts in combination with outcome labels through implicit process rewards to guide learning; and OpenReasoner-Zero (Hu et al., 2025), which represents a recent open-source implementation of RLVR methods. Moreover, we consider three popular approaches that combine SFT and RL: LUFFY (Yan et al., 2025): which integrates off-policy reasoning traces into RL training by combining them with on-policy rollouts; ReLIFT (Ma et al., 2025), which performs SFT intervention on hard samples;

and RL w/ SFT loss (Ma et al., 2025): which incorporates SFT loss into the GRPO objective. Additionally, models trained with either SFT or RL are included for comparison.

**Evaluation** Consistent with prior work (Ma et al., 2025; Yan et al., 2025), we evaluate all methods on five popular mathematical reasoning benchmarks, AIME-24, AIME-25, AMC (Li et al., 2024a), OlympiadBench (He et al., 2024), and MATH-500 (Hendrycks et al., 2021), as well as one OOD benchmark, MMLU-Pro (Wang et al., 2024a). Due to the relatively small test sets of AIME-24, AIME-25, and AMC, we report avg@32 as the evaluation metric, while avg@8 is used for OlympiadBench and MATH-500. To mitigate potential data contamination, multiple-choice options are randomly shuffled during evaluation.

**Main Results** The primary comparison results are presented in Table 1. Evaluated on five challenging math reasoning and one OOD benchmarks, **DALSA achieves a top overall accuracy of 54.3%, surpassing all compared baselines**. Specifically, DALSA surpasses the strongest RL baseline, Oat-Zero, by +8.4 percentage points in overall accuracy and consistently ranks within the top two across all individual benchmarks. When trained on the same dataset, DALSA outperforms standalone SFT and RL by +5.3 and +7.4 percentage points, respectively. Moreover, DALSA advances beyond all existing combination approaches by leveraging fine-grained, adaptive allocation of learning strategies and targeted enhancements for both RL and SFT. The impressive performance of LUFFY, ReLIFT, and DALSA further high-

*Table 2.* Overall performance on five competition-level mathematics benchmarks and one OOD benchmark, evaluated using the Qwen2.5-Math-1.5B and Qwen3-8B models.

| Method | AIME-24 | | AIME-25 | | AMC | | MATH-500 | | Olympiad | | MMLU-Pro | | Overall | |
|---|---|---|---|---|---|---|---|---|---|---|---|---|---|---|
| | *ACC* | *LEN* | *ACC* | *LEN* | *ACC* | *LEN* | *ACC* | *LEN* | *ACC* | *LEN* | *ACC* | *LEN* | *ACC* | *LEN* |
| *Qwen2.5-Math-1.5B** | 2.8 | 1971 | 1.3 | 2058 | 19.5 | 1533 | 45.4 | 1108 | 18.5 | 1619 | 5.2 | 1982 | 15.5 | 1712 |
| *Qwen2.5-Math-1.5B-Instruct** | 10.3 | 4042 | 7.6 | 3764 | 40.0 | 2809 | 77.8 | 1628 | 35.7 | 2834 | 33.7 | 4300 | 34.2 | 3230 |
| SFT* | 12.7 | 7925 | 13.0 | 7719 | 40.9 | 6953 | 71.8 | 5986 | 33.8 | 7231 | 23.5 | 5405 | 32.6 | 6870 |
| RL* | 9.8 | 2425 | 8.5 | 2195 | 44.1 | 1569 | 74.2 | 958 | 37.2 | 1695 | 31.4 | 1075 | 34.2 | 1653 |
| ReLIFT* | 14.3 | 3691 | 10.0 | 3416 | 47.2 | 2207 | 76.4 | 1308 | 39.6 | 2274 | 31.7 | 1724 | 36.5 | 2437 |
| DALSA (Ours) | **15.1** | 3440 | **13.2** | 3391 | **48.8** | 2115 | **77.6** | 1407 | **41.2** | 2098 | **33.2** | 1584 | **38.2** | 2339 |
| *Qwen3-8B* | 11.5 | 1754 | 9.2 | 1503 | 47.3 | 1121 | 60.9 | 734 | 27.4 | 1145 | 34.1 | 1001 | 31.7 | 1210 |
| SFT | 19.6 | 7824 | **24.1** | 7678 | 63.1 | 6690 | 78.2 | 4403 | 45.7 | 6476 | 56.0 | 4925 | 47.8 | 6333 |
| RL | 19.1 | 2409 | 20.8 | 1876 | 63.9 | 1893 | 77.2 | 1571 | 43.4 | 1911 | 58.0 | 1345 | 47.1 | 1834 |
| ReLIFT | 23.5 | 5893 | 21.6 | 5148 | 66.8 | 4067 | 79.5 | 2549 | 47.7 | 3927 | 57.9 | 2564 | 49.5 | 4025 |
| DALSA (Ours) | **24.9** | 5543 | 22.8 | 4897 | **68.3** | 3828 | **81.7** | 2242 | **49.2** | 3745 | **59.0** | 2301 | **51.0** | 3759 |

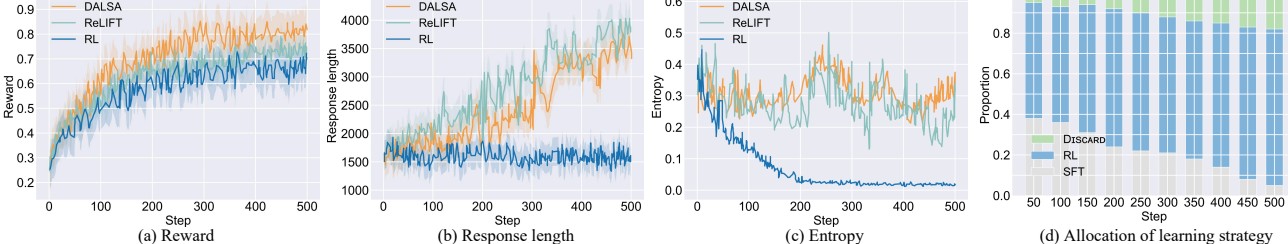

*Figure 2.* Training dynamics of rewards (a), response lengths (b), and training entropy (c), evaluated across RL, ReLIFT, and DALSA. (d) Variation in sample-wise allocation of learning strategies during the training process.

lights the efficacy of integrating SFT with RL, particularly in addressing challenging samples.

Beyond accuracy, we assess the efficiency of DALSA by examining the average response length (LEN) across all datasets. DALSA produces shorter solutions than SFT-related methods, reflecting more precise reasoning paths, while yielding longer lengths than RL training, indicating an improved ability to handle complex problems. These observations suggest that **DALSA not only improves accuracy but also reduces computational costs during inference**. Also, discussions in Appendix J show that DALSA requires less training time than previous integration approaches.

To further assess the generalization capability of DALSA, we extend its evaluation to additional LLMs, including Qwen2.5-Math-1.5B, Qwen3-8B, Qwen2.5-7B, and LLaMA3-8B. Table 2 summarizes the comparison results for Qwen2.5-Math-1.5B and Qwen3-8B, with others provided in Appendix F. In all cases, DALSA consistently surpasses standard RL and SFT baselines, as well as the advanced integration method ReLIFT, while maintaining reasonable response lengths. These findings highlight **DALSA's robustness and strong generalization across models of varying scales and capacities**.

**Training Analysis** Fig. 2 analyzes the training dynamics across multiple dimensions. As shown in Fig. 2(a), all methods exhibit increasing reward trends, reflecting progressive improvements in model performance. Notably, DALSA consistently achieves higher rewards and exhibits a faster convergence rate, highlighting its superior effectiveness in optimizing the objective. Regarding response lengths (Fig. 2(b)), RL tends to generate shorter outputs as training progresses, potentially indicating a reduced capacity to tackle more challenging questions. In contrast, DALSA demonstrates a gradual increase in response length, which remains lower on average than that of ReLIFT, reflecting an improved and more precise capability in handling complex cases. As illustrated in Fig. 2(c), the entropy of RL steadily declines over training, exhibiting reduced exploration. Conversely, both DALSA and ReLIFT preserve relatively high entropy levels, enabling models to continually discover novel solutions. Furthermore, Fig. 2(d) demonstrates that as training progresses, the proportion of difficult samples, i.e., those prioritized for SFT training, gradually decreases, reflecting the model's progressively improving capabilities. Further analyses regarding the allocator's training dynamics are presented in Appendix I.

*Table 3.* Performance comparison of training algorithms under the PEFT paradigm for eight commonsense reasoning tasks. LLaMA3-8B and LLaMA2-7B models are utilized, with LoRA and DoRA serving as the PEFT approaches.

| PEFT | Method | BoolQ | PIQA | SIQA | HellaSwag | WinoGrande | ARC-e | ARC-c | OBQA | Overall |
|------|--------|-------|------|------|-----------|------------|-------|-------|------|---------|
| | | | | | *LLaMA3-8B* | | | | | |
| LoRA | SFT | 70.8 | 85.2 | 79.9 | 91.7 | 84.3 | 84.2 | 71.2 | 79.0 | 80.8 |
| | RL | 70.1 | 84.1 | 80.2 | 86.6 | 81.9 | 80.7 | 66.2 | 77.6 | 78.4 |
| | ReLIFT | 71.8 | 85.9 | 81.1 | 91.2 | 84.4 | 86.1 | 74.7 | 80.9 | 82.0 |
| | DALSA (Ours) | **73.8** | **87.7** | **82.2** | **92.9** | **85.8** | **87.1** | **76.5** | **82.8** | **83.6** |
| DoRA | SFT | 74.6 | 89.3 | 79.9 | 95.5 | 85.6 | 90.5 | 80.4 | 85.8 | 85.2 |
| | RL | 73.5 | 88.6 | 79.1 | 95.1 | 84.8 | 89.9 | 79.7 | 84.8 | 84.4 |
| | ReLIFT | 74.6 | 89.7 | 80.7 | 95.3 | 86.1 | 91.1 | 81.5 | 86.9 | 85.7 |
| | DALSA (Ours) | **76.9** | **91.4** | **82.2** | **95.9** | **87.2** | **91.6** | **83.0** | **87.9** | **87.0** |
| | | | | | *LLaMA2-7B* | | | | | |
| LoRA | SFT | 69.8 | 79.9 | 79.5 | 83.6 | 82.6 | 79.8 | 64.7 | 81.0 | 77.6 |
| | RL | 68.6 | 79.1 | 79.4 | 77.6 | 79.0 | 76.8 | 61.2 | 76.6 | 74.8 |
| | ReLIFT | 67.9 | 84.1 | 80.4 | 88.5 | 82.3 | 83.1 | 69.1 | 80.7 | 79.5 |
| | DALSA (Ours) | **71.7** | **85.2** | **81.5** | **89.6** | **84.2** | **84.3** | **70.2** | **83.3** | **81.3** |
| DoRA | SFT | 71.8 | 83.7 | 76.0 | 89.1 | 82.6 | 83.7 | 68.2 | 82.4 | 79.7 |
| | RL | 71.0 | 82.6 | 74.7 | 88.0 | 81.7 | 82.9 | 67.0 | 81.6 | 78.7 |
| | ReLIFT | 73.8 | 84.9 | 76.9 | 90.3 | 83.8 | 84.5 | 70.4 | 83.6 | 81.0 |
| | DALSA (Ours) | **75.1** | **85.7** | **79.0** | **90.9** | **84.8** | **85.2** | **71.5** | **84.4** | **82.1** |

*Table 4.* Ablation studies to assess the contributions of the different components of the proposed DALSA framework. These studies include an evaluation of the learning strategy selection from the three options, SFT, RL, and DISCARD, as well as an analysis of the effectiveness of anti-curriculum weighting and the ALS loss function.

| Method | AIME-24 | AIME-25 | AMC | MATH-500 | Olympiad | MMLU-Pro | Overall |
|--------|---------|---------|-----|----------|----------|----------|---------|
| DALSA | **29.4** | 26.0 | **66.6** | **88.5** | **59.8** | **55.5** | **54.3** |
| w/o SFT | 23.5 | 21.8 | 62.7 | 86.6 | 49.5 | 48.0 | 48.7 |
| w/o RL | 27.5 | **27.8** | 60.4 | 86.2 | 55.1 | 46.2 | 50.6 |
| w/o DISCARD | 28.9 | 25.3 | 66.2 | 88.1 | 59.5 | 54.9 | 53.8 |
| w/o Anti-curriculum weighting | 28.4 | 25.3 | 66.1 | 88.0 | 59.4 | 55.0 | 53.7 |
| w/o ALS loss | 28.5 | 25.7 | 66.1 | 88.2 | 59.4 | 55.1 | 53.8 |

## 4.2. Parameter-Efficient Fine-Tuning

We also assess the effectiveness of DALSA under a PEFT setting, conducting experiments on commonsense reasoning, mathematical reasoning, and code generation tasks. Due to space constraints, only the results for commonsense reasoning and math reasoning tasks are presented here, while others are presented in Appendix H.

**Setups** We fine-tune LLaMA2-7B (Touvron et al., 2023) and LLaMA3-8B (Grattafiori et al., 2024) on the Commonsense170k dataset (Hu et al., 2023) and evaluate them across eight commonsense reasoning benchmarks: BoolQ (Clark et al., 2019), PIQA (Bisk et al., 2020), SIQA (Sap et al., 2019), HellaSwag (Zellers et al., 2019), WinoGrande (Sakaguchi et al., 2020), ARC-e (Clark et al., 2018), ARC-c, and OBQA (Mihaylov et al., 2018). All tasks are formulated as multiple-choice questions, with performance reported in terms of overall accuracy. Moreover, we fine-tune three

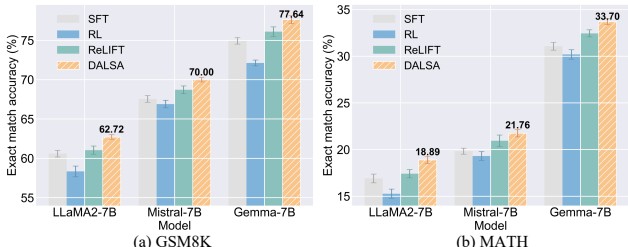

*Figure 3.* Performance comparison on two mathematical reasoning tasks across three LLMs using LoRA as the PEFT method.

models, LLaMA2-7B, Mistral-7B (Jiang et al., 2023), and Gemma-7B (Mesnard et al., 2024), on the MetaMathQA dataset (Yu et al., 2024), which comprises 395k samples. For evaluation, we employ the test sets of both GSM8K and MATH, reporting the exact match accuracy against the ground truth. The LoRA rank is set to 32 for commonsense

*Table 5.* Further ablation studies to evaluate the effectiveness of the anti-curriculum weighting scheme and the ALS loss.

| Method | AIME-24 | AIME-25 | AMC | MATH-500 | Olympiad | MMLU-Pro | Overall |
|---|---|---|---|---|---|---|---|
| RL* | 21.1 | 17.5 | 62.1 | 85.7 | 48.6 | 46.3 | 46.9 |
| SFT* | 26.9 | 25.5 | 59.8 | 84.8 | 52.6 | 44.4 | 49.0 |
| Anti-Curriculum RL | 22.4 | 18.0 | **63.1** | **86.6** | 50.3 | **47.5** | 48.2 |
| SFT with ALS Loss | **27.6** | **25.8** | 63.0 | 85.9 | **54.1** | 45.7 | **50.4** |

reasoning and 64 for math reasoning tasks, with additional hyperparameter details provided in Appendix G.

**Main Results**   Table 3 provides a comprehensive comparison of SFT, RL, ReLIFT, and DALSA across eight common-sense reasoning tasks using the LLaMA3-8B and LLaMA2-7B models. DALSA consistently achieves the highest performance across different models and datasets, yielding an average improvement of +1.5% over the strongest baseline. In addition, Fig. 3 presents the results on two mathematical reasoning benchmarks, where DALSA outperforms all competing methods. We further assess DALSA on larger LLMs, including LLaMA3-70B and Mixtral-8×7B. As shown in Fig. 6(c) of the Appendix, DALSA consistently and substantially surpasses SFT, RL, and the advanced ReLIFT method, even at these larger model scales. Together, these results demonstrate **the efficacy of the proposed DALSA training framework, which adaptively allocates the most suitable learning strategy to each individual sample, thereby enhancing overall model performance**.

### 4.3. Ablation Studies

To evaluate the contributions of the various components in our framework, we conduct a series of ablation experiments. Specifically, we assess the performance of our framework under different configurations. In one scenario, we remove SFT so that the learning strategy for the samples is solely selected from RL and DISCARD. In another configuration, we eliminate RL, and the learning strategy is chosen exclusively from SFT and DISCARD. Additionally, we explore the case where DISCARD is removed, with the learning strategy being selected only from SFT and RL. We also compare the performance of the standard CE loss with that of the ALS loss, and finally, we substituted the anti-curriculum RL algorithm with the standard RL approach. The results in Table 4 demonstrate that each component of DALSA, including the selection among SFT, RL, and DISCARD strategies, as well as the incorporation of the anti-curriculum weighting scheme and the ALS loss, contributes to the overall performance gains. This confirms that every component is essential and none of them is redundant. Furthermore, we compare the effectiveness of the two learning strategies enhanced by the directional improvements we proposed, relative to the vanilla learning strategy. As shown in Table 5,

the targeted improvement scheme we proposed is effective for both the SFT and RL learning strategies.

## 5. Conclusion

This study presents DALSA, a tailored training framework for LLMs that dynamically assigns learning strategies to samples based on their unique training characteristics. It first extracts sample-specific learning dynamics, such as perplexity and entropy, and inputs them into a strategy allocator, which adaptively determines the optimal training strategy, SFT, RL, or DISCARD, for each sample. Moreover, targeted enhancements, including anti-curriculum weighting for RL and ALS loss for SFT, are incorporated to further improve training effectiveness. Across both full fine-tuning and PEFT settings, DALSA consistently outperforms existing baselines in improving LLMs' reasoning performance.

## Acknowledgments

This work was supported by NSFC under Grant 625B2009.

## Impact Statement

This study presents work aimed at advancing the field of LLM training approaches. There are many potential societal consequences of our work, none of which we feel must be specifically highlighted here.

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

## A. Preliminary Experiments

Prior work (Ma et al., 2025; Qiu et al., 2025; Li et al., 2025b) validated that SFT excels on samples beyond model knowledge, whereas RL better refines responses within its competence. This motivates our use of difficulty-aware strategies. We conduct additional experiments using ReLIFT-style difficulty thresholds: zero-accuracy samples as hard, accuracy >0.9 as easy, and the rest as medium. Model performance is then evaluated under three settings. Setting I: SFT for hard, RL for medium, and easy samples discarded. Setting II: RL for hard, SFT for medium, easy samples discarded. Setting III: SFT for hard, RL for medium and easy. From results presented in Table 6, Setting I achieves the highest performance, validating our motivation.

## B. Theoretical Analysis of Anti-Curriculum Reinforcement Learning

In this section, we provide a theoretical analysis of the proposed anti-curriculum RL algorithm. Our analysis focuses on nonconvex finite-sum optimization problems of the form:

$$\max_{\boldsymbol{\theta} \in \mathbb{R}^d} \mathcal{J}^{\mathrm{RL}}(\boldsymbol{\theta}) := \frac{1}{n} \sum_{i=1}^{n} \mathcal{J}_i^{\mathrm{RL}}(\boldsymbol{\theta}), \tag{12}$$

where both $\mathcal{J}^{\mathrm{RL}}$ and each $\mathcal{J}_i^{\mathrm{RL}}$ ($i \in [n]$) are nonconvex. We denote the class of such finite-sum, Lipschitz-smooth functions by $\mathcal{J}_n$. Here, the optimization is performed over functions in $\mathcal{J}_n$ using a hard-sample–prioritized weighted policy gradient estimator.

### B.1. Policy Gradient Formulation

The standard policy gradient objective seeks to maximize the expected advantage $\mathcal{A}$:

$$\mathcal{J}^{\mathrm{RL}}(\boldsymbol{\theta}) = \mathbb{E}_{\tau \sim \pi_{\boldsymbol{\theta}}}[\mathcal{A}(\tau)] \approx \frac{1}{n} \sum_{i=1}^{n} \mathcal{A}(\tau_i). \tag{13}$$

By the policy gradient theorem (Sutton & Barto, 2015; Zang et al., 2025), the gradient estimator is:

$$\nabla \mathcal{J}^{\mathrm{RL}}(\boldsymbol{\theta}) = \mathbb{E}_{\tau \sim \pi_{\boldsymbol{\theta}}}[\nabla \log \pi_{\boldsymbol{\theta}}(\tau) \cdot \mathcal{A}(\tau)] \approx \frac{1}{n} \sum_{i=1}^{n} \nabla \log \pi_{\boldsymbol{\theta}}(\tau_i) \cdot \mathcal{A}(\tau_i). \tag{14}$$

In our approach, each sample is assigned an anti-curriculum weight $w_i$, resulting in a weighted policy gradient estimator:

$$\tilde{\nabla} \mathcal{J}_i(\boldsymbol{\theta}) = w_i \cdot \nabla \log \pi_{\boldsymbol{\theta}}(\tau_i) \cdot \mathcal{A}(\tau_i), \tag{15}$$

$$\tilde{\nabla} \mathcal{J}(\boldsymbol{\theta}) = \mathbb{E}_{\tau \sim \pi_{\boldsymbol{\theta}}}[w(\tau) \cdot \nabla \log \pi_{\boldsymbol{\theta}}(\tau) \cdot \mathcal{A}(\tau)] \approx \frac{1}{n} \sum_{i=1}^{n} w_i \cdot \nabla \mathcal{J}_i^{\mathrm{RL}}(\boldsymbol{\theta}). \tag{16}$$

Under stochastic gradient ascent, the parameter update at iteration $k$ is:

$$\boldsymbol{\theta}^{k+1} = \boldsymbol{\theta}^k + \eta_k w_{i_k} \nabla \mathcal{J}_{i_k}^{\mathrm{RL}}(\boldsymbol{\theta}^k), \quad i \in [n], \tag{17}$$

where $\eta_k$ is the learning rate at iteration $k$, and $w_{i_k}$ is the weight assigned to sample $i$.

*Table 6.* Performance comparison under different settings of strategy allocation.

| Setting | AIME-24 | AIME-25 | AMC | MATH-500 | Olympiad | MMLU-Pro | Overall |
|---------|---------|---------|------|----------|----------|----------|---------|
| I | **28.8** | **24.2** | **65.9** | **88.2** | **58.8** | **54.5** | **53.4** |
| II | 26.1 | 22.9 | 62.6 | 85.8 | 54.1 | 52.0 | 50.6 |
| III | 28.3 | 24.1 | 65.3 | 87.8 | 58.6 | 54.1 | 53.0 |

## B.2. Definitions

To facilitate the analysis, we introduce the following definitions ([Yan et al., 2025](); [Huang et al., 2025]()):

**Definition B.1** (Lipschitz Smoothness). A function $\mathcal{J}^{\mathrm{RL}} : \mathbb{R}^d \to \mathbb{R}$ is *L-smooth* if there exists $L > 0$ such that

$$\left\| \nabla \mathcal{J}^{\mathrm{RL}}(\boldsymbol{\theta}') - \nabla \mathcal{J}^{\mathrm{RL}}(\boldsymbol{\theta}) \right\| \leq L \left\| \boldsymbol{\theta}' - \boldsymbol{\theta} \right\|, \quad \forall \boldsymbol{\theta}', \boldsymbol{\theta} \in \mathbb{R}^d. \tag{18}$$

**Definition B.2** ($\epsilon$-Accuracy). A point $\boldsymbol{\theta} \in \mathbb{R}^d$ is *$\epsilon$-accurate* if $\left\| \nabla \mathcal{J}^{\mathrm{RL}}(\boldsymbol{\theta}) \right\|^2 \leq \epsilon$. A stochastic iterative algorithm achieves $\epsilon$-accuracy after $k$ iterations if

$$\mathbb{E}\left[ \left\| \nabla \mathcal{J}^{\mathrm{RL}}(\boldsymbol{\theta}^k) \right\|^2 \right] \leq \epsilon, \tag{19}$$

where the expectation is taken over the algorithm's inherent stochasticity.

**Definition B.3** ($\sigma$-Bounded Gradients). $\mathcal{J}^{\mathrm{RL}} \in \mathcal{J}_n$ has $\sigma$-bounded gradients if for all $i \in [n]$ and $\boldsymbol{\theta} \in \mathbb{R}^d$,

$$\left\| \nabla \mathcal{J}_i^{\mathrm{RL}}(\boldsymbol{\theta}) \right\| \leq \sigma. \tag{20}$$

**Definition B.4** (Bounded Weights). The positive instance weights $w_i$ satisfy $\underline{w} \leq w_i \leq \overline{w}$ for constants $\underline{w}, \overline{w}$ and all $i \in [n]$.

**Definition B.5** (Positive Correlation with Difficulty). The anti-curriculum weights $w$ are proportional to the difficulty scores $\rho$. This assumption is introduced for theoretical analysis and is empirically approximated in practice through the proposed sample difficulty estimation method and the corresponding weighting function.

**Definition B.6** (Sample Difficulty). We define the sample difficulty $\rho_i$ as a measure that is positively correlated with the projection of the sample gradient onto the expected gradient direction $\left\langle \nabla \mathcal{J}^{\mathrm{RL}}(\boldsymbol{\theta}), \nabla \mathcal{J}_i^{\mathrm{RL}}(\boldsymbol{\theta}) \right\rangle$. The underlying rationale is that under-learned, high-difficulty samples tend to produce gradients of large magnitude, which are generally aligned with the expected gradient direction under near noise-free conditions. This definition is supported by the experimental findings in Section I.

## B.3. Proofs

We present the following results to capture the convergence outcomes of the anti-curriculum RL strategy.

**Theorem B.7.** *The objective function $\mathcal{J}^{\mathrm{RL}} \in \mathcal{J}_n$ of the policy gradient algorithm is a finite-sum Lipschitz-smooth function with constant $L$. The gradient of $\mathcal{J}^{\mathrm{RL}}$ is uniformly bounded by $\sigma$, and the positive weights $w$ are constrained within the interval $[\underline{w}, \overline{w}]$. Let the learning rate be defined as $\eta_k = \eta = c/\sqrt{K}$, where $c = \sqrt{\frac{2(\mathcal{J}^{\mathrm{RL}}(\boldsymbol{\theta}^*) - \mathcal{J}^{\mathrm{RL}}(\boldsymbol{\theta}^0))}{L\sigma^2 \overline{w}^2}}$ and $K$ is the total number of iterations. Moreover, $\rho$ denotes the sample difficulty, and $\boldsymbol{\theta}^*$ is an optimal solution. Then, the iterates of the anti-curriculum RL algorithm satisfy*

$$\min_{0 \leq k \leq K} \mathbb{E}\left[ \left\| \nabla \mathcal{J}^{\mathrm{RL}}\left(\boldsymbol{\theta}^k\right) \right\|^2 \right] \leq \sqrt{\frac{2\left(\mathcal{J}^{\mathrm{RL}}\left(\boldsymbol{\theta}^*\right) - \mathcal{J}^{\mathrm{RL}}\left(\boldsymbol{\theta}^0\right)\right) L \overline{w}^2}{K \underline{w}^2}} \sigma - \omega, \tag{21}$$

*where $\omega = \min_k \frac{C_k}{\overline{w}} \cdot \mathrm{Cov}\left(w_k, \rho_k\right)$, and $C_k$ is a positive constant specific to $k$. The anti-curriculum weights satisfy $\mathrm{Cov}\left(w_k, \rho_k\right) > 0$. Consequently, the weighted optimizer is stabilized and converges to a stationary point at a rate of $\mathcal{O}(1/\sqrt{K})$. Moreover, this bound is tighter than those achieved when the weights are independent of, or negatively correlated with, sample difficulty, indicating accelerated convergence in terms of iteration complexity.*

*Proof.* By the Lipschitz continuity of $\nabla \mathcal{J}^{\mathrm{RL}}$, the iterates generated by our algorithm satisfy the following inequality:

$$\mathbb{E}\left[ \mathcal{J}^{\mathrm{RL}}\left(\boldsymbol{\theta}^{k+1}\right) \right] \geq \mathbb{E}\left[ \mathcal{J}^{\mathrm{RL}}\left(\boldsymbol{\theta}^k\right) + \left\langle \nabla \mathcal{J}^{\mathrm{RL}}\left(\boldsymbol{\theta}^k\right), \boldsymbol{\theta}^{k+1} - \boldsymbol{\theta}^k \right\rangle - \frac{L}{2}\left\| \boldsymbol{\theta}^{k+1} - \boldsymbol{\theta}^k \right\|^2 \right]. \tag{22}$$

Substituting Eq. (17) into Eq. (22) yields:

$$
\begin{aligned}
\mathbb{E}\left[ \mathcal{J}^{\mathrm{RL}}\left(\boldsymbol{\theta}^{k+1}\right) \right] &\geq \mathbb{E}\left[ \mathcal{J}^{\mathrm{RL}}\left(\boldsymbol{\theta}^k\right) \right] + \eta_k \mathbb{E}\left[ \left\langle \nabla \mathcal{J}^{\mathrm{RL}}\left(\boldsymbol{\theta}^k\right), w_{i_k} \nabla \mathcal{J}_{i_k}^{\mathrm{RL}}\left(\boldsymbol{\theta}^k\right) \right\rangle \right] - \frac{L\eta_k^2}{2} \mathbb{E}\left[ w_{i_k}^2 \left\| \nabla \mathcal{J}_{i_k}^{\mathrm{RL}}\left(\boldsymbol{\theta}^k\right) \right\|^2 \right] \\
&\geq \mathbb{E}\left[ \mathcal{J}^{\mathrm{RL}}\left(\boldsymbol{\theta}^k\right) \right] + \eta_k \mathbb{E}\left[ \left\langle \nabla \mathcal{J}^{\mathrm{RL}}\left(\boldsymbol{\theta}^k\right), w_{i_k} \nabla \mathcal{J}_{i_k}^{\mathrm{RL}}\left(\boldsymbol{\theta}^k\right) \right\rangle \right] - \frac{L\eta_k^2 \overline{w}^2}{2} \sigma^2.
\end{aligned} \tag{23}
$$

The second inequality is derived from the gradient and weight boundedness assumptions stated in Definitions B.3 and B.4. Focusing on the second term on the right-hand side of Eq. (23), we have

$$
\begin{aligned}
\mathbb{E}_{i_k}\left[\left\langle \nabla\mathcal{J}^{\mathrm{RL}}\left(\boldsymbol{\theta}^k\right), w_{i_k}\nabla\mathcal{J}_{i_k}^{\mathrm{RL}}\left(\boldsymbol{\theta}^k\right)\right\rangle\right] &= \left\langle \nabla\mathcal{J}^{\mathrm{RL}}\left(\boldsymbol{\theta}^k\right), \mathbb{E}_{i_k}\left[w_{i_k}\nabla\mathcal{J}_{i_k}^{\mathrm{RL}}\left(\boldsymbol{\theta}^k\right)\right]\right\rangle \\
&= \left\langle \nabla\mathcal{J}^{\mathrm{RL}}\left(\boldsymbol{\theta}^k\right), \mathbb{E}\left[w_{i_k}\right]\mathbb{E}_{i_k}\left[\nabla\mathcal{J}_{i_k}^{\mathrm{RL}}\left(\boldsymbol{\theta}^k\right)\right] + \mathrm{Cov}\left(w_{i_k}, \nabla\mathcal{J}_{i_k}^{\mathrm{RL}}\left(\boldsymbol{\theta}^k\right)\right)\right\rangle \\
&= \mathbb{E}[w_{i_k}]\|\nabla\mathcal{J}^{\mathrm{RL}}(\boldsymbol{\theta}^k)\|^2 + \langle\nabla\mathcal{J}^{\mathrm{RL}}(\boldsymbol{\theta}^k), \mathrm{Cov}(w_{i_k}, \nabla\mathcal{J}_{i_k}^{\mathrm{RL}}(\boldsymbol{\theta}^k))\rangle \\
&= \mathbb{E}\left[w_{i_k}\right]\left\|\nabla\mathcal{J}^{\mathrm{RL}}\left(\boldsymbol{\theta}^k\right)\right\|^2 + \mathrm{Cov}\left(w_{i_k}, \left\langle\nabla\mathcal{J}^{\mathrm{RL}}\left(\boldsymbol{\theta}^k\right), \nabla\mathcal{J}_{i_k}^{\mathrm{RL}}\left(\boldsymbol{\theta}^k\right)\right\rangle\right).
\end{aligned}
\tag{24}
$$

Based on the assumptions on the anti-curriculum weighting scheme and the definition of sample difficulty (Definitions B.5 and B.6), there exists a positive constant $C_{i_k} > 0$ such that

$$
\mathrm{Cov}\left(w_{i_k}, \left\langle\nabla\mathcal{J}^{\mathrm{RL}}\left(\boldsymbol{\theta}^k\right), \nabla\mathcal{J}_{i_k}^{\mathrm{RL}}\left(\boldsymbol{\theta}^k\right)\right\rangle\right) \geq C_{i_k}\cdot\mathrm{Cov}\left(w_{i_k}, \rho_{i_k}\right) > 0.
\tag{25}
$$

Subsequently, we define $C_k\mathrm{Cov}(w_k, \rho_k) = \min_i C_{i_k}\mathrm{Cov}(w_{i_k}, \rho_{i_k}) > 0$ and substitute Eqs. (24) and (25) into Eq. (23). By rearranging the formula, we then obtain

$$
\mathbb{E}\left[\left\|\nabla\mathcal{J}^{\mathrm{RL}}\left(\boldsymbol{\theta}^k\right)\right\|^2\right] \leq \frac{1}{\mathbb{E}\left[w_k\right]\eta_k}\mathbb{E}\left[\mathcal{J}^{\mathrm{RL}}\left(\boldsymbol{\theta}^{k+1}\right) - \mathcal{J}^{\mathrm{RL}}\left(\boldsymbol{\theta}^k\right)\right] - \frac{C_k\mathrm{Cov}\left(w_k, \rho_k\right)}{\mathbb{E}\left[w_k\right]} + \frac{L\eta_k\overline{w}^2}{2\mathbb{E}\left[w_k\right]}\sigma^2.
\tag{26}
$$

Summing Eq. (26) over $k = 0$ to $K-1$ and noting that the step size $\eta_k$ is constant ($\eta_k = \eta$), we obtain:

$$
\begin{aligned}
\min_{0\leq k\leq K}\mathbb{E}\left[\left\|\nabla\mathcal{J}^{\mathrm{RL}}\left(\boldsymbol{\theta}^k\right)\right\|^2\right] &\leq \frac{1}{K}\sum_{k=0}^{K-1}\mathbb{E}\left[\left\|\nabla\mathcal{J}^{\mathrm{RL}}\left(\boldsymbol{\theta}^k\right)\right\|^2\right] \\
&\leq \frac{1}{K}\sum_{k=0}^{K-1}\frac{1}{\eta\mathbb{E}[w_k]}\mathbb{E}\left[\mathcal{J}^{\mathrm{RL}}\left(\boldsymbol{\theta}^{k+1}\right) - \mathcal{J}^{\mathrm{RL}}\left(\boldsymbol{\theta}^k\right)\right] + \frac{1}{K}\sum_{k=0}^{K-1}\left[\frac{L\eta\overline{w}^2}{2\mathbb{E}[w_k]}\sigma^2 - \frac{C_k\cdot\mathrm{Cov}(w_k, \rho_k)}{\mathbb{E}[w_k]}\right] \\
&\leq \frac{1}{K\eta\underline{w}}\left(\mathcal{J}^{\mathrm{RL}}\left(\boldsymbol{\theta}^K\right) - \mathcal{J}^{\mathrm{RL}}\left(\boldsymbol{\theta}^0\right)\right) + \frac{L\eta\overline{w}^2}{2\underline{w}}\sigma^2 - \omega \\
&\leq \frac{1}{K\eta\underline{w}}\left(\mathcal{J}^{\mathrm{RL}}\left(\boldsymbol{\theta}^*\right) - \mathcal{J}^{\mathrm{RL}}\left(\boldsymbol{\theta}^0\right)\right) + \frac{L\eta\overline{w}^2}{2\underline{w}}\sigma^2 - \omega \\
&\leq \frac{1}{\sqrt{K}}\left(\frac{1}{c\underline{w}}\left(\mathcal{J}^{\mathrm{RL}}\left(\boldsymbol{\theta}^*\right) - \mathcal{J}^{\mathrm{RL}}\left(\boldsymbol{\theta}^0\right)\right) + \frac{Lc\overline{w}^2}{2\underline{w}}\sigma^2\right) - \omega,
\end{aligned}
\tag{27}
$$

where $\omega = \min_k \frac{C_k}{\overline{w}}\cdot\mathrm{Cov}\left(w_k, \rho_k\right) > 0$. The first inequality follows from the fact that the minimum is always less than or equal to the average. The second inequality is derived directly from Eq. (26). The third inequality arises from the boundedness of the weights, as stated in Definition B.4. The fourth inequality arises from the observation that $\mathcal{J}^{\mathrm{RL}}\left(\boldsymbol{\theta}^*\right) \geq \mathcal{J}^{\mathrm{RL}}\left(\boldsymbol{\theta}^K\right)$. Finally, the last inequality is obtained by substituting $\eta = c/\sqrt{K}$. By setting

$$
c = \sqrt{\frac{2\left(\mathcal{J}^{\mathrm{RL}}\left(\boldsymbol{\theta}^*\right) - \mathcal{J}^{\mathrm{RL}}\left(\boldsymbol{\theta}^0\right)\right)}{L\sigma^2\overline{w}^2}}
\tag{28}
$$

in the above formula, we obtain the desired result.

According to Theorem B.7, the policy gradient estimator with anti-curriculum weights is stabilized and converges to a stationary point at a rate of $\mathcal{O}(1/\sqrt{K})$, with a smaller convergence bound. In contrast, when the weights are not assigned based on a hard-sample prioritization scheme, i.e., when they are either independent of sample difficulty or biased toward easier samples, the additional term $\omega$ in the bound becomes non-positive. As a result, the bound becomes looser compared to that achieved under the anti-curriculum scheme, and convergence acceleration cannot be guaranteed.

$\square$

---

**Chat Template**

Your task is to follow a systematic, thorough reasoning process before providing the final solution. This involves analyzing, summarizing, exploring, reassessing, and refining your thought process through multiple iterations. Structure your response into two sections: **Thought** and **Solution**.

In the **Thought** section, present your reasoning using the format: "<think>\n {thoughts} </think>\n". Each thought should include detailed analysis, brainstorming, verification, and refinement of ideas.

After "</think>\n," in the **Solution** section, provide the final, logical, and accurate answer, clearly derived from the exploration in the Thought section.

If applicable, include the **Answer** in \boxed{} for closed-form results like multiple choices or mathematical solutions.

---

User: This is the problem: {Question}
Assistant: <think>

*Figure 4.* Illustration for the utilized chat template, which was adapted from Ma et al. (2025).

## C. Theoretical Analysis of Adversarial Label Smoothing Loss

As introduced in the main text, the proposed ALS loss is defined as follows:

$$\mathcal{L}_{\text{ALS}} = -\sum_{i=1}^{|\mathcal{V}|} \tilde{y}_i \log p_i, \quad \tilde{y}_i = (1-\vartheta)\, y_i^{\text{ref}} + \vartheta \delta_i, \tag{29}$$

where $\delta_i = \frac{p_i^{-\tau}}{\sum_v p_v^{-\tau}}$. Subsequently, we analyze the gradient of the logit vector $z_i$ for the ALS loss. Since $\tilde{y}_i$ depends on $p_i$, the gradient must be computed using the chain rule:

$$\frac{\partial \mathcal{L}_{\text{ALS}}}{\partial z_i} = \sum_{j=1}^{|\mathcal{V}|} \frac{\partial \mathcal{L}_{\text{ALS}}}{\partial p_j} \frac{\partial p_j}{\partial z_i} + \sum_{j=1}^{|\mathcal{V}|} \frac{\partial \mathcal{L}_{\text{ALS}}}{\partial \tilde{y}_j} \sum_{k=1}^{|\mathcal{V}|} \frac{\partial \tilde{y}_j}{\partial p_k} \frac{\partial p_k}{\partial z_i}. \tag{30}$$

By expanding step by step, we obtain

$$\frac{\partial \mathcal{L}_{\text{ALS}}}{\partial z_i} = \underbrace{p_i - \tilde{y}_i}_{\text{Standard CE gradient}} + \underbrace{\left\{ -\sum_{j=1}^{|\mathcal{V}|} (\log p_j)\, \vartheta \sum_{k=1}^{|\mathcal{V}|} \frac{\partial \delta_j}{\partial p_k} \frac{\partial p_k}{\partial z_i} \right\}}_{\text{Chain rule term}}, \tag{31}$$

where the chain rule term represents the correction of the logits with respect to the adversarial perturbation:

$$\frac{\partial \delta_j}{\partial p_k} = -\tau \delta_j \left( \frac{\mathbb{I}(j=k)}{p_j} - \frac{\delta_k}{p_k} \right), \quad \frac{\partial p_k}{\partial z_i} = p_k \left( \mathbb{I}(k=i) - p_i \right), \tag{32}$$

where $\mathbb{I}(j=k) = \begin{cases} 1 & \text{if } j = k \\ 0 & \text{if } j \neq k \end{cases}$.

Based on the decomposition above, the standard CE gradient drives the model predictions toward the smoothed labels, effectively reducing overfitting to one-hot targets. In contrast to conventional label smoothing using a fixed uniform distribution, ALS introduces an adaptive form of regularization that is more targeted and effective. By preventing the model's outputs from becoming extreme, it guides the model to allocate a meaningful probability mass to difficult-to-classify classes. Consequently, the predicted probability distribution becomes smoother, which can also enhance confidence calibration. Moreover, the chain term captures the second-order dependency of the loss on the logits through the probability-dependent

smoothed labels, thereby amplifying gradient updates for less confident predictions and enhancing the model's ability to learn from complex or marginal samples. This chain term also propagates gradients to non-target classes, mitigating extreme biases in the logits and promoting more stable generalization. Furthermore, the proposed ALS loss enables the adjustment of both the chain term strength and the sharpness of the adversarial perturbations through the hyperparameters $\vartheta$ and $\tau$, providing a flexible trade-off between training stability and regularization effectiveness.

Additionally, analogous to applying adversarial perturbations in the input space, the ALS loss can be interpreted as introducing adversarial perturbations in the label space, as shown below:

$$\tilde{\boldsymbol{y}} = \boldsymbol{y}^{\mathrm{ref}} + \hat{\boldsymbol{\delta}}, \quad \hat{\boldsymbol{\delta}} = \arg\max_{\boldsymbol{\delta}} \mathcal{L}\left(f_{\boldsymbol{\theta}}\left(\boldsymbol{x}\right), \boldsymbol{y}^{\mathrm{ref}} + \boldsymbol{\delta}\right). \tag{33}$$

This can be viewed as a distribution-level (Chen et al., 2026) adversarial training, which encourages the model to preserve correct classification decisions even when the target labels are subject to slight uncertainty.

## D. Application of Straight-Through Estimator

In the forward pass, the allocator produces a probability distribution $\boldsymbol{\mathcal{S}} = [s^{\mathrm{SFT}}, s^{\mathrm{RL}}, s^{\mathrm{D}}]$ based on the sample features $\boldsymbol{\xi}$. A discrete decision is then made via the $\arg\max$ operation:

$$\begin{aligned}
\alpha &= \mathbb{I}\{\arg\max\_k \in \{\mathrm{SFT}, \mathrm{RL}, \mathrm{D}\} s^k = \mathrm{SFT}\}, \\
\beta &= \mathbb{I}\{\arg\max\_k \in \{\mathrm{SFT}, \mathrm{RL}, \mathrm{D}\} s^k = \mathrm{RL}\}.
\end{aligned} \tag{34}$$

To enable the propagation of gradients, we introduce the following operations in implementation:

$$\begin{aligned}
\alpha &= \left(\mathbb{I}\{\arg\max_{k \in \{\mathrm{SFT}, \mathrm{RL}, \mathrm{D}\}} s^k = \mathrm{SFT}\} - \mathrm{s}^{\mathrm{SFT}}\right).\mathrm{detach}() + s^{\mathrm{SFT}}, \\
\beta &= \left(\mathbb{I}\{\arg\max_{k \in \{\mathrm{SFT}, \mathrm{RL}, \mathrm{D}\}} s^k = \mathrm{RL}\} - \mathrm{s}^{\mathrm{RL}}\right).\mathrm{detach}() + s^{\mathrm{RL}}.
\end{aligned} \tag{35}$$

This ensures that during the forward pass, the $\alpha$ and $\beta$ values are exactly discrete.

During the backward pass, STE treats the discrete $\arg\max$ operation as an identity function, i.e.,

$$\frac{\partial \alpha}{\partial s^{\mathrm{SFT}}} \approx 1, \quad \frac{\partial \beta}{\partial s^{\mathrm{RL}}} \approx 1.$$

Accordingly, the gradient with respect to the allocator parameters $\boldsymbol{\Omega}$ is computed as:

$$\nabla\_{\boldsymbol{\Omega}}\mathcal{J} = \mathbb{E}\_{\boldsymbol{x} \sim \mathcal{B}}\left[\frac{\partial \mathcal{J}}{\partial \alpha\_{\boldsymbol{x}}} \cdot \frac{\partial \alpha\_{\boldsymbol{x}}}{\partial s\_{\boldsymbol{x}}^{\mathrm{SFT}}} \cdot \frac{\partial s\_{\boldsymbol{x}}^{\mathrm{SFT}}}{\partial \boldsymbol{\Omega}} + \frac{\partial \mathcal{J}}{\partial \beta\_{\boldsymbol{x}}} \cdot \frac{\partial \beta\_{\boldsymbol{x}}}{\partial s\_{\boldsymbol{x}}^{\mathrm{RL}}} \cdot \frac{\partial s\_{\boldsymbol{x}}^{\mathrm{RL}}}{\partial \boldsymbol{\Omega}}\right].$$

With the STE approximation, this expression simplifies to

$$\nabla\_{\boldsymbol{\Omega}}\mathcal{J} \approx \mathbb{E}_{\boldsymbol{x} \sim \mathcal{B}}\left[\mathcal{J}\_{\boldsymbol{x}}^{\mathrm{SFT}} \cdot \nabla\_{\boldsymbol{\Omega}} s\_{\boldsymbol{x}}^{\mathrm{SFT}} + \mathcal{J}\_{\boldsymbol{x}}^{\mathrm{RL}} \cdot \nabla\_{\boldsymbol{\Omega}} s\_{\boldsymbol{x}}^{\mathrm{RL}}\right].$$

In this way, gradients can be smoothly propagated back to the allocator.

## E. More Experimental Details for Full-Tuning

### E.1. Dataset Details

**OpenR1-Math-220k**[6] (Bakouch et al., 2025) is a large-scale dataset specifically curated for evaluating mathematical reasoning capabilities. It comprises 220k problems spanning diverse domains such as algebra, geometry, number theory, and calculus. Each problem is accompanied by two to four reasoning chains generated by the DeepSeek-R1[7] (Guo et al., 2025) model. These reasoning chains have been rigorously validated using verification tools, including Math Verify[8] and

---

[6] https://huggingface.co/datasets/open-r1/OpenR1-Math-220k
[7] https://github.com/deepseek-ai/DeepSeek-R1
[8] https://github.com/huggingface/Math-Verify

---

### LLaMA Chat Template

Question: {Question}.
Answer: Let's think step by step.

---

*Figure 5.* Illustration of the chat template employed for the LLaMA model, adapted from Ma et al. (2025).

*Table 7.* Overall performance on five competition-level mathematics benchmarks and one OOD benchmark, evaluated using the Qwen2.5-7B and LLaMA3-8B models.

| Method | AIME-24 | | AIME-25 | | AMC | | MATH-500 | | Olympiad | | MMLU-Pro | | Overall | |
|---|---|---|---|---|---|---|---|---|---|---|---|---|---|---|
| | *ACC* | *LEN* | *ACC* | *LEN* | *ACC* | *LEN* | *ACC* | *LEN* | *ACC* | *LEN* | *ACC* | *LEN* | *ACC* | *LEN* |
| *Qwen2.5-7B** | 6.2 | 1632 | 2.4 | 1370 | 31.1 | 1076 | 63.8 | 617 | 26.5 | 1068 | 32.8 | 945 | 27.1 | 1118 |
| *Qwen2.5-7B-Instruct** | 11.5 | 1939 | 6.1 | 1500 | 41.0 | 1629 | 73.0 | 1753 | 37.3 | 1550 | 53.1 | 3325 | 37.0 | 1949 |
| SFT* | 15.7 | 7786 | **18.6** | 7533 | 49.8 | 6424 | 80.8 | 4252 | 44.3 | 6234 | 54.6 | 4869 | 44.0 | 6183 |
| RL* | 15.5 | 1784 | 13.0 | 1487 | 50.4 | 1422 | 80.0 | 917 | 42.2 | 1438 | 57.6 | 954 | 43.1 | 1334 |
| ReLIFT* | 19.1 | 5522 | 14.7 | 4831 | 51.9 | 3792 | 81.6 | 2211 | 46.1 | 3649 | 56.4 | 2173 | 45.0 | 3696 |
| DALSA (Ours) | **20.6** | 5089 | **16.3** | 4694 | **53.4** | 3524 | **84.1** | 2036 | **47.0** | 3341 | **58.3** | 2005 | **46.6** | 3448 |
| *LLaMA3-8B-Instruct** | 5.6 | 1396 | 0.7 | 1401 | 20.7 | 1012 | 44.0 | 622 | 13.8 | 1063 | 38.1 | 357 | 20.5 | 975 |
| SFT* | 0.8 | 2033 | **1.5** | 2032 | 11.5 | 2007 | 28.6 | 1903 | 8.6 | 1997 | 28.2 | 1558 | 13.2 | 1922 |
| RL* | 1.8 | 940 | 0.0 | 909 | 10.8 | 797 | 28.2 | 616 | 7.3 | 790 | 39.4 | 549 | 14.6 | 767 |
| ReLIFT* | 1.3 | 1236 | 0.2 | 1272 | 11.9 | 1048 | 35.2 | 741 | 11.0 | 1050 | 44.2 | 650 | 17.3 | 1000 |
| DALSA (Ours) | **2.4** | 1197 | 1.1 | 1108 | **12.9** | 1025 | **37.8** | 766 | **12.8** | 907 | **46.3** | 608 | **18.9** | 935 |

LLaMA3-70B-Instruct (Grattafiori et al., 2024), ensuring that each problem contains at least one correct solution path. By providing multiple verified reasoning traces per problem, this dataset presents a substantial challenge for models, requiring them not only to arrive at correct answers but also to comprehend, reproduce, and generalize complex multi-step reasoning processes across various mathematical contexts.

**AIME-24**[9] is a benchmark dataset derived from the 2024 American Invitational Mathematics Examination (AIME), a highly regarded high school mathematics competition in the United States. The dataset is designed to evaluate a model's capability (Jiang et al., 2025a; Cai et al., 2025) to solve challenging mathematical problems by generating detailed step-by-step solutions alongside the correct answers. AIME-24 is provided in JSONL format, with each line representing a complete problem instance. The problems span a range of mathematical domains, including elementary algebra, geometry, trigonometry, number theory, probability, and combinatorics. Each examination comprises 15 problems, each with an integer answer in the range [0, 999], requiring sophisticated reasoning and advanced problem-solving skills. By encompassing both diverse topics and high-complexity problem-solving, AIME-24 serves as a rigorous benchmark for assessing models' mathematical reasoning and stepwise solution generation abilities.

**AIME-25**[10] directly utilizes the officially released problems from the 2025 AIME Part I and Part II as the evaluation dataset. AIME itself is a high-difficulty mathematics competition for high school students, targeting top performers who have excelled in the American Mathematics Competitions (AMC). The problems cover key mathematical domains (Yu et al., 2026; Rao et al., 2025a), including algebra, geometry, number theory, and combinatorics. The evaluation protocol is both concise and rigorous: LLMs are required to independently solve these problems in the same manner as human participants, producing a single integer answer in the range [0, 999].

**AMC**[11] (Li et al., 2024a) is a validation dataset comprising problems from AMC, specifically the AMC12 exams from 2022 and 2023. In total, the dataset includes 83 problems, all sourced from AMC12 2022 and 2023, with problem content extracted from the AoPS wiki. The AMC12 exam is a 25-question, 75-minute multiple-choice competition intended for students in grades 10 and below. It covers foundational mathematical topics, including elementary algebra, basic geometry, area and volume calculations, elementary number theory, and introductory probability. This dataset functions as an internal

---

[9] https://huggingface.co/datasets/HuggingFaceH4/aime_2024
[10] https://huggingface.co/datasets/math-ai/aime25
[11] https://artofproblemsolving.com/wiki/index.php/AMC_12_Problems_and_Solutions

validation set, providing a collection of competition-level problems that are comparable in difficulty to those found in the AMC12 and AIME examinations, thereby enabling robust evaluation of models' mathematical reasoning capabilities.

**MATH-500**[12] (Hendrycks et al., 2021) is a carefully curated evaluation subset designed to rigorously assess mathematical reasoning capabilities. It consists of 500 randomly sampled problems drawn from the MATH dataset[13] (Hendrycks et al., 2021). The subset was constructed by OpenAI in late 2024, following the incorporation of approximately 90% of the original 5k MATH problems into the training data for RL on the O1-series models. MATH-500 preserves the diversity and complexity of the original benchmark while providing a clean evaluation set, thereby mitigating potential data contamination and ensuring reliable assessment of model performance.

**Olympiad**[14] (He et al., 2024) refers to datasets comprising mathematical olympiad-level problems, representing some of the most challenging tasks in mathematical reasoning. Unlike existing benchmarks related to olympiad problems, these datasets are exclusively focused on mathematics and consist of extensive collections of competition-level questions. The problems are systematically organized into more than 33 sub-domains and span over 10 distinct difficulty tiers. Solving these problems requires exceptional mathematical insight, creativity, and advanced problem-solving techniques typically exhibited in international mathematics competitions.

**MMLU-Pro** (Wang et al., 2024a) is an enhanced extension of the original MMLU[15] benchmark, developed to provide a more rigorous and challenging evaluation of language understanding capabilities. Building on MMLU, which encompasses 57 subjects spanning domains such as mathematics, history, law, and medicine, MMLU-Pro is specifically designed to increase task difficulty and reduce potential shortcuts while preserving the broad coverage across diverse academic disciplines. This benchmark not only evaluates models' factual knowledge but also their ability to apply this knowledge in contextually nuanced scenarios, thereby offering a more robust and comprehensive assessment of professional-level language understanding.

### E.2. Experimental Settings

To ensure a fair comparison, we utilize 8 rollouts per prompt with a fixed learning rate of $1 \times 10^{-6}$, and training is conducted on $8 \times$ A800 GPUs. Following Ma et al. (2025), all methods, except for the RL approach applied after SFT, are trained for 600 steps on Qwen2.5-Math-7B, whereas the remaining base models are trained for 500 steps. Evaluations are performed using the vLLM[16] (Kwon et al., 2023) inference framework, with a decoding temperature fixed at 0.6 and a maximum generation length capped at 8,192 tokens. For baseline comparisons, the configurations of all competing methods are aligned with those reported in the ReLIFT study (Ma et al., 2025). To improve computational efficiency, asynchronous updates are employed, wherein the learning strategy allocator is updated once for every five updates of the LLM. A layer normalization layer is integrated into the learning strategy allocator in our framework to ensure stability during training. Moreover, the temperature parameter $\psi$ in the $\psi$-Softmax function is fixed at 0.3. Whenever available, results from ReLIFT are directly adopted; otherwise, the performance of these methods is reproduced under the corresponding experimental settings.

To enhance the reasoning capabilities of the base model, we adopt the complex chat template proposed by Ma et al. (2025), as illustrated in Fig. 4. For the LLaMA model, however, we do not use the aforementioned system prompt, as empirical results indicate that the model does not reliably follow such instructions. Instead, a simplified prompt is employed, as shown in Fig. 5. Furthermore, to evaluate the effectiveness of our approach, we adopt a straightforward reward function $\mathcal{R}$ that is consistently applied across all training procedures:

$$\mathcal{R} = \begin{cases} 1, & \text{if the answer is correct,} \\ 0, & \text{otherwise.} \end{cases} \tag{36}$$

## F. Additional Experimental Results for Full Fine-Tuning

Beyond the full fine-tuning experiments presented in the main text, we also conduct similar experiments on the Qwen2.5-7B (Yang et al., 2024a) and LLaMA3-8B (Grattafiori et al., 2024) models to further assess the adaptability of the proposed

---

[12]https://huggingface.co/datasets/PrimeIntellect/MATH-500
[13]https://huggingface.co/datasets/qwedsacf/competition_math
[14]https://github.com/OpenBMB/OlympiadBench
[15]https://huggingface.co/datasets/cais/mmlu
[16]https://github.com/vllm-project/vllm

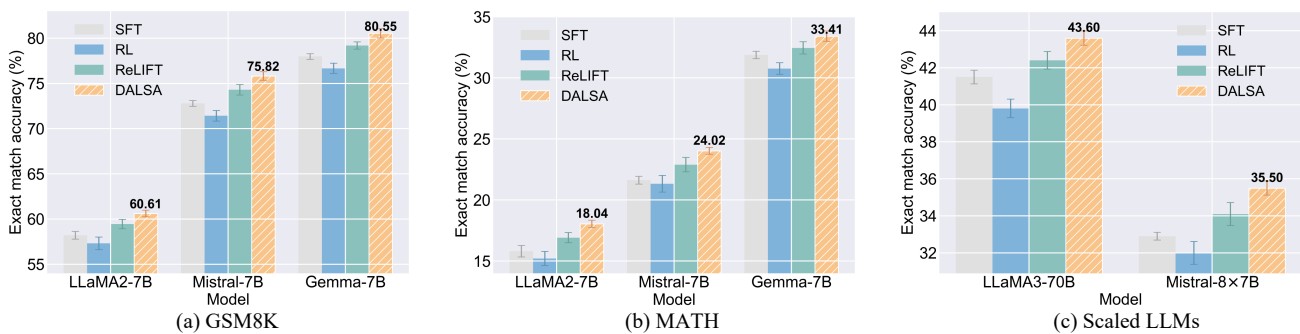

*Figure 6.* (a) and (b): Performance comparison achieved by different methods on two mathematical reasoning tasks using PiSSA as the PEFT approach. (c): Performance comparison on scaled models evaluated on the MATH dataset with LoRA as the PEFT method.

*Table 8.* Evaluation results for coding proficiency using the HumanEval and MBPP datasets, with the value of rank set to 64.

| Method | LoRA | | PiSSA | | Overall |
|---|---|---|---|---|---|
| | **HumanEval** | **MBPP** | **HumanEval** | **MBPP** | |
| *LLaMA2-7B* | | | | | |
| SFT | 18.18±0.35 | 35.40±0.21 | 22.06±0.43 | 37.15±0.37 | 28.20 |
| RL | 18.01±0.68 | 35.23±0.51 | 21.01±0.44 | 36.24±0.50 | 27.62 |
| ReLIFT | 19.65±0.50 | 37.27±0.67 | 23.58±0.33 | 38.64±0.48 | 29.79 |
| DALSA (Ours) | **21.10**±0.28 | **38.02**±0.45 | **25.06**±0.31 | **39.99**±0.53 | **31.04** |
| *Mistral-7B* | | | | | |
| SFT | 43.87±0.21 | 58.28±0.40 | 46.97±0.29 | 62.72±0.38 | 52.96 |
| RL | 40.92±0.59 | 56.45±0.48 | 45.12±0.63 | 61.99±0.36 | 51.12 |
| ReLIFT | 45.21±0.54 | 60.13±0.37 | 49.11±0.46 | 64.84±0.60 | 54.82 |
| DALSA (Ours) | **46.60**±0.30 | **61.97**±0.29 | **50.73**±0.48 | **66.12**±0.55 | **56.35** |
| *Gemma-7B* | | | | | |
| SFT | 53.71±0.42 | 65.55±0.33 | 54.34±0.27 | 66.21±0.28 | 59.95 |
| RL | 52.66±0.63 | 65.48±0.52 | 53.12±0.59 | 65.07±0.48 | 59.08 |
| ReLIFT | 54.23±0.37 | 66.41±0.45 | 55.67±0.49 | 67.35±0.52 | 60.92 |
| DALSA (Ours) | **55.54**±0.35 | **67.78**±0.29 | **56.74**±0.31 | **68.41**±0.47 | **62.12** |

DALSA framework. Regarding the LLaMA model, the full OpenR1-Math-46k[17] dataset was proven too complex for the LLaMA3-8B model (Ma et al., 2025). Consequently, we adopt the smaller and more manageable subset constructed by Ma et al. (2025) for evaluation. Specifically, 11k samples were sampled from OpenR1[18] (Bakouch et al., 2025) where DeepSeek-R1 (Guo et al., 2025) provided the correct solution, and the solution length was constrained to under 2,048 tokens. SFT was then carried out with a learning rate of $2 \times 10^{-5}$, following the recommendation of Liu et al. (2025b). For both RL training and evaluation, the maximum response length was set to 2,048 tokens. The experimental results, shown in Table 7, reveal that DALSA achieved a 1.6% improvement over the best baseline. This outcome demonstrates that the proposed approach, through tailored learning strategy allocation and the integration of enhanced SFT and RL strategies, delivers superior performance, a benefit that is consistent across different model architectures.

## G. More Experimental Details for Parameter-Efficient Fine-Tuning

The hyperparameter configurations for PEFT are detailed as follows. The rank is set to 32 for commonsense reasoning tasks and 64 for math reasoning and code generation tasks, while the scaling factor in LoRA[19] is chosen from {64, 128}. A

---

[17] https://huggingface.co/datasets/Elliott/Openr1-Math-46k-8192
[18] https://github.com/huggingface/open-r1
[19] https://github.com/microsoft/LoRA

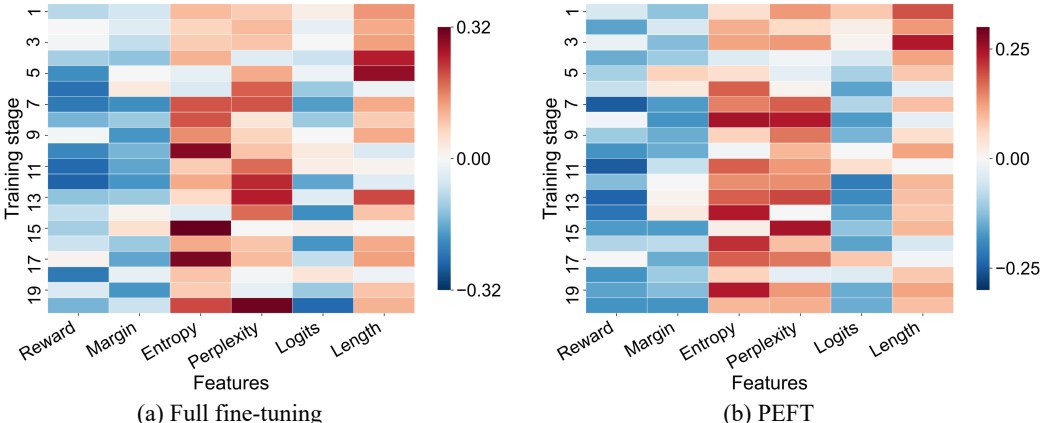

*Figure 7.* Evolution of Pearson correlation coefficients between sample difficulty and various training characteristics throughout the training process.

dropout rate of $0.05$ is applied during training. We adopt AdamW as the optimizer, with a learning rate of $3 \times 10^{-4}$ for SFT, in conjunction with a linear learning rate scheduler. The batch size is fixed at $64$. The parameter-efficient modules are integrated into the query, key, value, up, and down projection layers. The learning rate for the RL component is also set to $3 \times 10^{-4}$, and the RL-based methods are implemented using the GRPO optimization framework. All other configurations follow those described in Section 4.1.

## H. More Experiments for Parameter-Efficient Fine-Tuning

### H.1. Math Reasoning Tasks

The results on two mathematical reasoning benchmarks, GSM8K and MATH, using LoRA as the PEFT method and evaluated across three LLMs (Yang et al., 2026a), LLaMA2-7B (Touvron et al., 2023), Mistral-7B (Jiang et al., 2023), and Gemma-7B (Mesnard et al., 2024), are presented in the main text (Fig. 3). Here, we provide additional results for these settings using PiSSA (Meng et al., 2024) as the PEFT approach. As shown in Figs. 6(a) and (b), the proposed DALSA framework achieves the best performance. In addition, we assess DALSA on larger-scale models, including LLaMA3-70B and Mixtral-8×7B, where the LoRA rank for these models is set to 4. Fig. 6(c) reports the results on the MATH dataset, demonstrating that DALSA consistently and substantially outperforms SFT, RL, and the advanced ReLIFT method, even at larger model scales.

### H.2. Code Generation Tasks

We fine-tune three LLMs, LLaMA2-7B, Mistral-7B, and Gemma-7B, on the CodeFeedback dataset (Zheng et al., 2024), using LoRA and PiSSA as the PEFT approaches. The resulting models are then evaluated on code generation performance using the HumanEval (Chen et al., 2021) and MBPP (Austin et al., 2021) benchmarks. As shown in Table 8, the proposed DALSA framework consistently outperforms SFT, RL, and ReLIFT across various coding tasks and LLMs, irrespective of the PEFT method employed.

## I. Analysis for Training Dynamics of Learning Strategy Allocator

As described in the main text, we adopt the probability of a sample being allocated to SFT training as a proxy for sample difficulty. Based on this definition, we investigate the evolution of the Pearson correlation coefficients between the proposed difficulty measure and various input training characteristics of the learning strategy allocator throughout the training process. For visualization purposes, we divide the training process into 20 stages, with each stage consisting of 25 training steps. As illustrated in Fig. 7, a general positive correlation is observed between sample difficulty and entropy, perplexity, and input length. In contrast, the reward mean, predicted margin, and logits norm show an overall negative correlation with sample difficulty. This aligns with intuitive reasoning, where increased information entropy and perplexity indicate greater prediction uncertainty, suggesting higher learning difficulty. Consequently, larger values of these two indicators are associated with

*Table 9.* Comparison of computational complexity across different SFT-RL integration approaches.

| Method | GPU hours | Overall performance |
|---|---|---|
| RL w/ SFT loss | $113.5 \times 8$ | 48.6 |
| LUFFY | $73 \times 8$ | 50.9 |
| ReLIFT | $52 \times 8$ | 52.6 |
| DALSA | $45.5 \times 8$ | **54.3** |

*Table 10.* Ablation study results for the effectiveness of the dynamic and adaptive strategy allocation.

| Method | AIME-24 | AIME-25 | AMC | MATH-500 | Olympiad | MMLU-Pro | Overall |
|---|---|---|---|---|---|---|---|
| Frozen | 28.9 | 25.1 | 66.0 | 88.1 | 59.0 | 54.7 | 53.6 |
| Static | 28.5 | 24.0 | 65.6 | 88.2 | 58.1 | 54.2 | 53.1 |
| DALSA | **29.4** | **26.0** | **66.6** | **88.5** | **59.8** | **55.5** | **54.3** |

more challenging samples. Additionally, longer sequence lengths tend to correlate with greater difficulty. On the other hand, a lower reward mean suggests poorer performance, a smaller predicted margin typically reflects higher model uncertainty, and a reduced logits norm implies poorer model fitting. Thus, smaller values of these three characteristics are indicative of greater learning difficulty. Given that our framework significantly outperforms prior methods, this suggests that our approach, which integrates a combination of training characteristics to assess sample difficulty, offers an accurate and effective assessment.

## J. Time Complexity

An analysis of the required GPU hours for various approaches, as shown in Table 9, demonstrates that DALSA outperforms other integration methods by achieving superior performance with fewer training hours. The efficiency of DALSA stems from the fact that the extracted training characteristics can be derived by reusing intermediate computations generated during the standard training process. Moreover, the allocator is implemented as a lightweight two-layer MLP with fewer than 0.04 million parameters, resulting in a negligible computational overhead compared to that of LLMs. These findings underscore DALSA's ability to optimize both computational resources and model performance effectively.

## K. More Sensitivity Analyses

This section investigates the sensitivity of DALSA to several key hyperparameters introduced. Specifically, both the anti-curriculum weighting strategy and the ALS loss involve two hyperparameters. Encouragingly, our experimental results demonstrate that these hyperparameters exhibit strong generalizability across different tasks and therefore do not require extensive tuning in practice.

For the proposed anti-curriculum weighting function, the hyperparameters include $\lambda_{\max}$, which controls the degree of the hard-first scheme, and $k$, which determines the growth rate of $\lambda$. For the ALS loss, the hyperparameters include $\vartheta$, which controls the extent of label smoothing, and $\tau$, which governs the sharpness of the adversarial label perturbation. As illustrated in Fig. 8, the optimal performance is achieved when $\lambda_{\max} = 0.8$ and $k = 0.5$. Moreover, the model performance remains stable over a broad range of $\vartheta \in [0.003, 0.007]$ and $\tau \in [0.25, 0.75]$, indicating the stable effectiveness of the proposed DALSA framework to hyperparameter variations.

## L. More Ablation Studies

The core motivation for alternating optimization is to establish a dynamic feedback loop between model training and strategy allocation, where the allocator is updated to reflect the evolving optimization landscape and prioritize the most beneficial strategies. In contrast, static partitioning or a frozen allocator assumes constant difficulty and cannot accommodate progressive dynamics. To validate this, we conduct two ablations: 1) Frozen allocator: Pretrained for 200 steps and then fixed. 2) Static partitioning: Each sample is assigned (offline) a fixed strategy based on initial LLM states using a trained allocator. From results presented in Table 10, our max–min optimization achieves the best performance, highlighting the

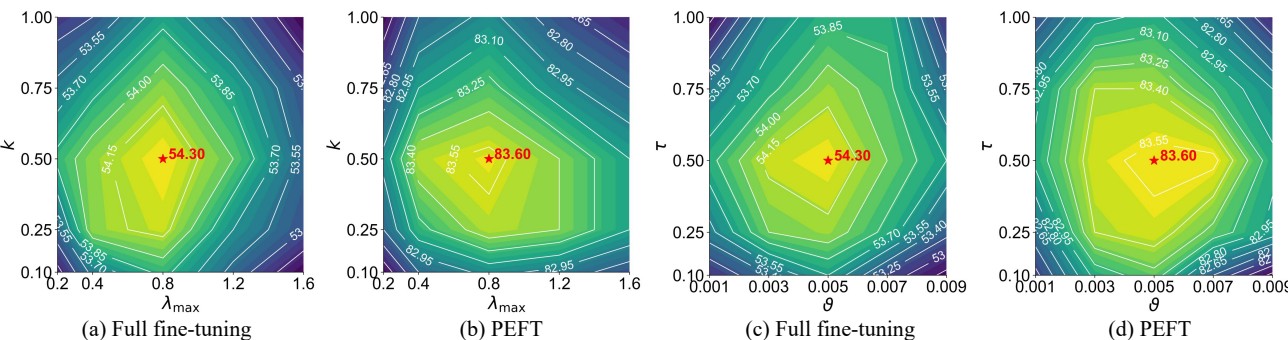

*Figure 8.* (a) and (b): Sensitivity analyses of the hyperparameters in the anti-curriculum weighting strategy, including the values of $\lambda_{max}$ and $k$. (c) and (d): Sensitivity analyses of the hyperparameters in the ALS loss function, including the values of $\vartheta$ and $\tau$.

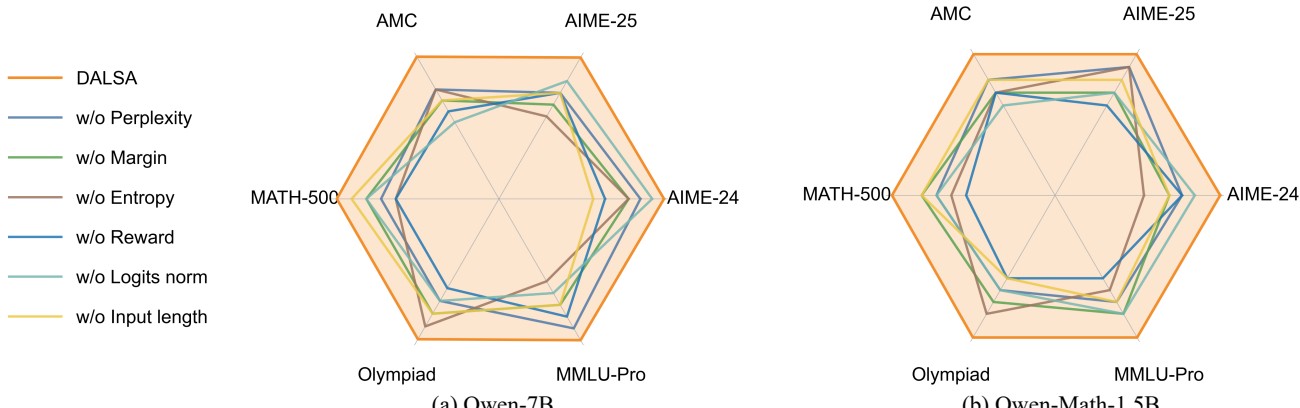

*Figure 9.* Ablation studies for the input training characteristics of the learning strategy allocator.

importance of dynamic allocation.

## M. Effectiveness of Various Training Characteristics

We conduct ablation experiments to assess the impact of different inputs to the learning strategy allocator. Specifically, we systematically remove one characteristic at a time and analyze the resulting changes in the framework's performance. The results of these experiments are presented in Fig. 9, which indicate that each training characteristic plays a crucial role in capturing the training dynamics and learning difficulty of the samples. As a result, employing all six features as inputs to the learning strategy allocator yields the best performance. This observation also motivates us to consider incorporating additional aspects of sample training features in future work, with the goal of achieving a more comprehensive understanding of sample training dynamics and a more robust model of sample learning difficulty.

## N. Case Study

We present a case study examining the assignment of samples to different learning strategies. Tables 11, 12, and 13 present representative examples of samples allocated to the SFT, RL, and discarded categories, respectively. The results indicate that the most challenging samples are preferentially assigned to SFT-based training within our framework, while particularly simple samples are designated for discarding. These findings underscore the effectiveness and sound rationale of the proposed learning strategy allocation mechanism in DALSA.

*Table 11.* Examples of samples allocated for SFT-based training.

| Instance | Explanation |
| --- | --- |
| Consider a rectangular parallelepiped $ABCD - A'B'C'D'$, where $ABCD$ is the bottom face with the letters assigned in a clockwise direction, and $A, B, C$, and $D$ are directly below $A', B', C'$, and $D'$ respectively. The parallelepiped is divided into eight pieces by three planes that are orthogonal to each other and parallel to the faces of the parallelepiped. For each vertex $P$ of the parallelepiped, let $V_P$ denote the volume of the piece of the parallelepiped that contains $P$. Given that $V_A = 40, V_C = 300, V_{B'} = 360$ and $V_{C'} = 90$, what is the volume of the parallelepiped $ABCD - A'B'C'D'$? | Requires the model to perform precise geometric reasoning in three-dimensional space as well as algebraic computations, abilities that depend on a deep understanding of 3D structures, the relationships among subdivided volumes, and the proportional relationships between variables. |
| Find the number of integer solutions of the equation $x^{2016} + (2016! + 1!)x^{2015} + (2015! + 2!)x^{2014} + ... + (1! + 2016!) = 0$. | Involves a high-degree polynomial (up to $x^{2016}$) with extremely large factorial coefficients, making the direct determination of integer solutions require advanced number-theoretic knowledge and deep mathematical reasoning. |
| Find all functions $f : \mathbf{N}_+ \to \mathbf{N}_+$, such that for all positive integers $a, b$, there exists a non-degenerate triangle with side lengths $a, f(b), f(b + f(a) - 1)$ (a triangle is called non-degenerate if its three vertices are not collinear). | Constitutes a reasoning task that combines functional number theory with inequality constraints, requiring characterization of the global structure of the function under a universal quantification over all $a$ and $b$. |
| As shown in Figure 1, $S - ABC$ is a triangular pyramid with three edges mutually perpendicular, and $O$ is a point within the base $ABC$. If $\angle OSA = \alpha, \angle OSB = \beta, \angle OSC = \gamma$, then the range of $\tan\alpha \cdot \tan\beta \cdot \tan\gamma$ is ( ). (A) $[2\sqrt{2}, +\infty)$ (B) $(0, 2\sqrt{2})$ (C) $[1, 2\sqrt{2}]$ (D) $(1, 2\sqrt{2})$. | Involves integrated reasoning in spatial geometry and three-dimensional vector analysis, challenging the model's geometric intuition and requiring multi-step algebraic and trigonometric computations. |
| In a rectangular array of points, with 5 rows and $N$ columns, the points are numbered consecutively from left to right beginning with the top row. Thus, the top row is numbered 1 through $N$, the second row is numbered $N + 1$ through $2N$, and so forth. Five points, $P_1, P_2, P_3, P_4$, and $P_5$, are selected so that each $P_i$ is in row $i$. Let $x_i$ be the number associated with $P_i$. Now renumber the array consecutively from top to bottom, beginning with the first column. Let $y_i$ be the number associated with $P_i$ after the renumbering. It is found that $x_1 = y_2, x_2 = y_1, x_3 = y_4, x_4 = y_5$, and $x_5 = y_3$. Find the smallest possible value of $N$. | Requires the model to simultaneously track row and column relationships under a dual-numbering system in a two-dimensional matrix and to establish a multi-step algebraic mapping between row-major and column-major numbering, involving complex combinatorial and arithmetic reasoning. |
| The number $3^{2009}$ is represented as the sum of $k$ consecutive natural numbers. What is the greatest possible value of $k$? | Involves analyzing the number-theoretic structure of an exponentially large integer and requires a deep understanding of implicit mathematical properties such as factorization and parity constraints. |
| $p(x, y, z)$ is a polynomial with real coefficients such that: (1) $p(tx, ty, tz) = t2f(y - x, z - x)$ for all real $x, y, z, t$ (and some function $f$); (2) $p(1, 0, 0) = 4, p(0, 1, 0) = 5$, and $p(0, 0, 1) = 6$; and (3) $p(\alpha, \beta, \gamma) = 0$ for some complex numbers $\alpha, \beta, \gamma$ such that $|\beta - \alpha| = 10$. Find $|\gamma - \alpha|$. | Involves structural analysis of higher-order multivariate polynomials, integrated reasoning over real and complex variables, and deriving precise relationships among unknowns based on functional equations and linear scaling properties. |

*Table 12.* Examples of samples allocated for RL-based training.

| Instance | Explanation |
| --- | --- |
| Theorem on the lengths of a tangent and a secant; the product of the entire secant and its external part From a point $M$, located outside the circle at a distance of $\sqrt{7}$ from the center, a secant is drawn, the internal part of which is half the external part and equal to the radius of the circle. Find the radius of the circle. | Requires mastery of various geometric conditions, such as the fundamental formulas of the tangent-secant length theorem, an understanding of the distance from an external point to a circle, and the ability to translate geometric conditions into algebraic equations for multi-step calculations. |
| Six chairs sit in a row. Six people randomly seat themselves in the chairs. Each person randomly chooses either to set their feet on the floor, to cross their legs to the right, or to cross their legs to the left. There is only a problem if two people sitting next to each other have the person on the right crossing their legs to the left and the person on the left crossing their legs to the right. The probability that this will (or not) happen is given by $\frac{m}{n}$ where $m$ and $n$ are relatively prime positive integers. Find $m + n$. | Requires an understanding of the fundamental principles of probability and combinatorics, while also being able to analyze the "adjacent conflict" conditions in multi-step discrete events and compute the total number of conflict-free arrangements using either the exclusion principle or recursive methods. |
| The sides $a$ and $b$ of a rectangle change according to the laws $a = (2t + 1)$ cm, $b = (3t + 2)$ cm. At what rate is its area $S$ changing at the moment $t = 4$ s? | Involves knowledge of calculus and analysis of functional relationships, requiring an understanding of how the side lengths change over time and the ability to compute the derivative of the area. |
| Let $S = \{1, 2, \cdots, 2005\}$. If any set of $n$ pairwise coprime numbers in $S$ contains at least one prime number, find the minimum value of $n$. | Requires understanding coprime and prime numbers and determining the minimal $n$ via combinatorial number theory. |
| Points $A, B, C, D$ are on a line in the given order and do not coincide. The lengths of segments $AB$, $AC$, and $AD$ are $x, y$, and $z$ respectively. If segments $AB, CD$ can rotate around points $B, C$ respectively until points $A$ and $D$ coincide, forming a triangle with a definite area, then among the following three inequalities: I. $x < \frac{z}{2}$; II. $y < x + \frac{z}{2}$; III. $y < \frac{z}{2}$, which must be satisfied are (A) I only. (B) II only. (C) I and II only. (D) II and III only. (E) I, II, and III. | Involves a comprehensive analysis of geometric spatial transformations and inequality conditions, requiring the accurate assessment of area constraints after multiple rotational operations and the selection of the inequalities that are satisfied. |
| Four fair cubic dice are rolled once. The probability that at least three of the four dice show the same number is ( ). (A) $\frac{1}{36}$ (B) $\frac{7}{72}$ (C) $\frac{1}{9}$ (D) $\frac{5}{36}$ (E) $\frac{1}{6}$. | Involves fundamental probability concepts and combinatorial analysis, requiring the model to perform multi-step logical reasoning and systematic enumeration. |
| Four teams $A, B, C,$ and $D$ are participating in a football tournament. Each team plays exactly one match against each of the others, and teams are awarded 2, 1, or 0 "bonus points" for a win, draw, or loss, respectively. The day after the tournament concludes, Peter hears the end of a radio report: "...Team $D$ finished fourth. Thus, no two teams had the same point total. The match between $A$ and $B$ was the only one that ended in a draw." Peter is disappointed that his favorite team was not mentioned in this part of the report. Nevertheless, from the information he heard and his knowledge of the tournament format, he can determine not only the placement but also the point total of his favorite team. How is this possible? | Involves knowledge of discrete mathematics and combinatorial reasoning, requiring the accurate deduction of each team's point totals and the determination of the unique solution. |

*Table 13.* Examples of samples allocated for discarding.

| Instance | Explanation |
|---|---|
| When going from the first to the third floor, Petya walks 36 steps. When going from the first floor to his own floor in the same entrance, Vasya walks 72 steps. On which floor does Vasya live? | A straightforward proportional reasoning problem solvable via basic arithmetic. |
| Which of the following numbers is not an integer? A $\frac{2011}{1}$ B $\frac{2012}{2}$ C $\frac{2013}{3}$ D $\frac{2014}{4}$ E $\frac{2015}{5}$. | Only involves basic integer division and divisibility checking. |
| Person A and Person B play a "Guess the Number" game with a fair die (the six faces of the die are numbered $1, 2, \cdots, 6$). A and B each think of a number on the die, denoted as $a$ and $b$ respectively. If $\lvert a - b \rvert \leqslant 1$, it is said that "A and B are in sync". The probability that A and B are in sync is _____. | Involves only basic discrete probability calculations and combinatorial counting, without requiring complex reasoning. |
| Write the equation of the plane passing through point $A$ and perpendicular to vector $\overrightarrow{BC}$. $A(1; -1; 5)$ $B(0; 7; 8)$ $C(-1; 3; 8)$. | Only involves basic three-dimensional vector operations and the standard formula for a plane equation. |
| For what value of $a$ does the polynomial $P(x) = x^{1000} + ax^2 + 9$ divide by $x + 1$? | Involves basic algebraic operations and equation solving. |
| The members of a working group "Young Botanists" supported their patron LPG in fruit growing. To this end, they kept a 2.6 ha orchard free from pests, on which an average of 150 apple trees stood per hectare. Afterwards, an average of 50 kg of apples were harvested from each tree. Calculate how many tons of apples were harvested in total on the orchard under these conditions! | Only involves basic arithmetic operations. |
| For the imaginary number i $\left(i^2 = -1\right)$, consider the set $S = \left\{i, i^2, i^3, i^4\right\}$. It is easy to see that the product of any two elements in $S$ is still in $S$. Now, we define the multiplicative identity $\theta$ in $S$: for any $a \in S$, we have $a\theta = \theta a = a$. Then $\theta$ is ( ). (A) i (B) $i^2$ (C) $i^3$ (D) $i^4$. | Only involves the basic powers of the imaginary number i and the concept of the multiplicative identity. |
| The ratio between the number of men and women in the city of Campo Verde is $\frac{2}{3}$. The average age of men is 37 years and that of women is 42 years. What is the average age of the inhabitants of Campo Verde? | Only involves the basic concept of weighted averages. |
| The set includes 8 weights: 5 identical round, 2 identical triangular, and one rectangular weight weighing 90 grams. 1 round and 1 triangular weight balance 3 round weights. Additionally, 4 round weights and 1 triangular weight balance 1 triangular, 1 round, and 1 rectangular weight. How much does the triangular weight weigh? | Involves basic algebraic equation modeling and solving linear systems of equations. |
| Ted's grandfather exercised on a treadmill for three days, running 2 miles each day. It is known that the speeds on the first, second, and third days were 5 miles/hour, 3 miles/hour, and 4 miles/hour, respectively. If his grandfather maintained a speed of 4 miles/hour throughout, he would have spent ( ) minutes less on the treadmill. (A) 1 (B) 2 (C) 3 (D) 4 (E) 5. | Involves only basic arithmetic operations. |

