# OpenReview forum: "Tailoring the Training: Difficulty-Aware Learning Strategy Allocation for Large Language Models"
_ICML.cc/2026/Conference — ICML 2026 regular_

### Official Review · Reviewer_JSbB · 2026-03-09

**Soundness:** 3
**Presentation:** 1
**Significance:** 2
**Originality:** 3
**Overall Recommendation:** 3
**Confidence:** 4

**Summary:**

The paper proposes DALSA, a training framework that dynamically selects the most suitable learning strategy for each training sample when improving large language models. Instead of using fixed schedules to combine supervised fine-tuning (SFT) and reinforcement learning (RL), DALSA estimates sample difficulty using signals such as perplexity, entropy, prediction margins, and reward statistics, and employs a learned allocator to assign each sample to SFT, RL, or discard it if already mastered. The framework jointly trains the allocator and the model through alternating optimization, and introduces two additional techniques, anti-curriculum weighting to encourage RL to focus on harder samples and adversarial label smoothing to regularize SFT. Experiments across multiple LLMs and tasks show that this adaptive strategy allocation consistently improves reasoning performance over standard SFT, RL, and prior hybrid approaches.

**Compliance With Llm Reviewing Policy:**

Affirmed.

**Final Justification:**

The paper proposes a novel and interesting idea of difficulty-aware allocation of training strategies, and the experimental results are reasonably comprehensive. The rebuttal is helpful and addresses several of my concerns by providing additional analyses and ablations, which strengthens the empirical support of the method.

However, I remain concerned that the method introduces multiple additional components (e.g., ALS and various design choices), making it somewhat over-engineered with many hyperparameters, while some ablations show limited impact, raising questions about their necessity. In addition, the presentation could be improved, as important analyses are deferred to the appendix while less central results occupy the main text.

Overall, it is somewhat unfortunate that the paper’s presentation and design choices dilute what is otherwise a strong core idea. I recommend a final score of 3.

[Updated] I thank the authors for the detailed and patient follow-up. I acknowledge that many of the additional analyses, ablations, and clarifications are already available as discussed. However, without seeing a revised manuscript, it is difficult to fully assess how these improvements affect the overall presentation and clarity. Based on the current version, I maintain my score of 3, while agreeing that the work is interesting and has a valuable core idea.

**Key Questions For Authors:**

1. What is the computational cost of computing the training characteristics in Section 3.2? For example, the reward mean appears to require generation, which may introduce additional overhead. In addition, are all these characteristics necessary, and are there ablation studies evaluating their contributions?

2. The paper interprets a higher $s^{SFT}$ as indicating higher sample difficulty. Can the authors design experiments to empirically validate this assumption? Without such evidence, the interpretation seems insufficiently supported.

3. Is the time-dependent weighting function λ(t) in the anti-curriculum weighting necessary? It would be helpful to provide ablation results evaluating its impact.

4. I am particularly interested in the additional computational cost introduced by the method and the potential training time savings it may bring. Please clearly quantify both.

5. (Minor) Section 2 reads more like a related work discussion rather than a preliminary section.

**Limitations:**

No. The paper does not discuss the limitations; several concerns remain unaddressed, as noted in the weaknesses and questions.

**Strengths And Weaknesses:**

**Strengths:**

1. The core idea of automatically allocating training strategies (RL, SFT, or discard) based on sample difficulty is novel.

2. The experimental evaluation is relatively comprehensive, covering multiple models, tasks, and training settings, and demonstrates consistent overall performance improvements over existing baselines.

**Weaknesses:**

1. The method design is relatively complex, involves many additional hyperparameters, and lacks sufficient analysis. In particular, the allocator and the SFT/RL training are jointly optimized, which may lead to degenerate behaviors (e.g., the allocator exploiting shortcuts), yet the paper provides little analysis of the learned allocation strategy, making it difficult to assess the method’s validity.

2. The motivation for the adversarial label smoothing component is unclear. It appears more like a general SFT trick and is not directly related to the paper’s core idea of difficulty-aware strategy allocation. Its inclusion weakens the conceptual focus, and the lack of ablation makes it hard to judge its necessity. It is also unclear whether the baselines use the same technique, raising concerns about the fairness of the comparison.

3. The presentation and organization of the paper could be improved. The PEFT results in the main text seem less central and could be moved to the appendix, while more useful analyses would be additional efficiency evaluations (e.g., computational cost or training time savings) and deeper analysis of the proposed method.

---

> ### Author Rebuttal · Authors · 2026-03-30
>
> Dear Reviewer JSbB,
>
> We sincerely appreciate your thoughtful review and valuable comments. Thank you for recognizing the novelty of our research, the comprehensiveness of the experiments, and our strong performance. Below are our detailed responses to your concerns.
> >Q1: Method design.
>
> We humbly address your concerns as follows:
> - The risk of trivial shortcuts can be **mitigated by max–min alternating optimization** ($\max_{\boldsymbol{\theta}}\min_{\boldsymbol{\Omega}} \mathcal{J}$), which discourages degenerate assignments and drives selecting beneficial strategies.
> - **Fig. 2(d)** analyzes the evolution of strategy assignments, with the PEFT results for commonsense tasks below: SFT decreases while RL and DISCARD increase as model improves, indicating **the allocator learns a meaningful curriculum rather than collapsing**.
> |Step|100|200|300|400|500|
> |:-|:-:|:-:|:-:|:-:|:-:|
> |Discard (%)|8|11|14|16|19|
> |SFT|31|22|17|11|3|
> |RL|61|67|69|73|78|
> - Despite added components, the allocator is lightweight (**<0.04M**). Moreover, **Appendix K** presents sensitivity analyses that clearly identify recommended values and stable ranges for hyperparameters.
>
> >Q2: Adversarial label smoothing.
>
> DALSA is a difficulty-aware training framework, where an allocator assigns samples to SFT, RL, or DISCARD, and ALS and the anti-curriculum mechanism further refines pathway-specific optimization.
>
> ALS can mitigate overfitting in SFT and enhance robustness to label noise. Ablations in **Appendix L (Tables 9 and 10)** show that **removing ALS degrades performance, though DALSA w/o ALS still outperforms ReLIFT**.
> ||AIME-24|AIME-25|AMC|MATH-500|Olympiad|MMLU-Pro|Overall|
> |:-|:-:|:-:|:-:|:-:|:-:|:-:|:-:|
> |DALSA|29.4|26.0|66.6|88.5|59.8|55.5|**54.3**|
> |DALSA w/o ALS|28.5|25.7|66.1|88.2|59.4|55.1|53.8|
> |ReLIFT|28.3|22.9|65.1|87.9|57.3|53.9|52.6|
> >Q3: Organization.
>
> Following your valuable suggestion, we moved most PEFT experiments to Appendix and used the freed space for complexity analysis and component ablations.
> >Q4: Costs and contribution of characteristics.
>
> We respectfully respond to your concerns as follows:
> - Characteristic computation uses fewer forward passes than RL rollouts, enabling reuse for RL-trained samples. For samples with other strategies, it remains cheaper than ReLIFT, which performs full RL rollouts per sample. **Profiling average per-step runtime against ReLIFT**, with additional time normalized by ReLIFT’s total runtime, shows that feature extraction, involving simple vector operations, introduces **only 3%** overhead.
> ||LLM forward|Feature extraction|Allocator forward|LLM update|Allocator update|
> |:-|:-:|:-:|:-:|:-:|:-:|
> |ReLIFT|100%|-|-|100%|-|
> |DALSA|88%|3%|<0.1%|81%|<0.1%|
> - Ablations on extracted features (**Appendix M, Fig. 10**), with partial Qwen-7B results below, clearly **demonstrate their respective contributions**.
> ||AIME-24|AMC|
> |:-|:-:|:-:|
> |DALSA|**20.6**|**53.4**|
> |w/o reward|20.1|52.9|
> |w/o perplexity|20.4|53.1|
> |w/o margin|20.3|53.0|
> |w/o entropy|20.3|53.1|
> |w/o logits|20.5|52.8|
> |w/o length|20.0|53.0|
>
> >Q5: Higher $s^{\rm SFT}$ indicates higher difficulty.
>
> The following findings support this claim.
> - **Fig. 7** shows Pearson correlations between $s^{\rm SFT}$ and extracted characteristics. **Metrics that typically increase with difficulty (e.g., entropy, perplexity, and length) exhibit positive correlations, whereas others correlate negatively**.
> - **Tables 12–14** show cases for SFT, RL, and DISCARD, with their average probability distributions as follows. **SFT samples are harder and correspond to the highest $s^{\rm{SFT}}$ (0.94)**.
> |SFT|RL|Discard|
> |:-|:-:|:-:|
> |[0.94,0.05,0.01]|[0.05,0.92,0.03]|[0.02,0.06,0.92]|
> - We further partition samples by $s^{\rm SFT}$ and evaluate them with Qwen3-8B, showing that **accuracy decreases with increasing $s^{\rm{SFT}}$**.
> |Group|High $s^{\rm SFT}$|Medium|Low|
> |:-|:-:|:-:|:-:|
> |ACC|10.6|35.4|72.2|
>
> >Q6: Ablation for $\lambda(t)$.
>
> We respond to your concerns as follows:
> - Since a large fixed $\lambda$ early in training may overemphasize noises or outliers and destabilize optimization, we adopt a time-dependent $\lambda(t)$ to **first learn from a broad distribution and gradually focus on informative samples**.
> - Ablations comparing $\lambda(t)$ with fixed values ($\lambda\equiv0.1, 0.5, 0.8$) show that **$\lambda(t)$ consistently performs best**.
> ||BoolQ|PIQA|SIQA|Hell.|Wino.|ARC-e|ARC-c|OBQA|Overall|
> |:-|:-:|:-:|:-:|:-:|:-:|:-:|:-:|:-:|:-:|
> |$\lambda(t)$|73.8|87.7|82.2|92.9|85.8|87.1|76.5|82.8|**83.6**|
> |0.1|73.0|87.2|81.7|92.6|85.3|86.7|75.9|81.9|83.0|
> |0.5|73.3|87.3|81.9|92.6|85.6|86.9|75.8|82.2|83.2|
> |0.8|72.6|86.3|81.7|92.1|84.9|86.5|75.1|81.4|82.6|
>
> >Q7: Overhead.
>
> From **our response to Q4**, DALSA adds only ~3% overhead (feature extraction + allocator), while SFT and DISCARD reduce LLM forward/update to 88%/81%, cutting total cost by 12.5% (**Table 7**).
> >Q8: Title of Section 2.
>
> We have renamed Section 2 from *Preliminary* to *Related Work*.

---

> > ### Author Rebuttal · Reviewer_JSbB · 2026-04-02
> >
> > The rebuttal addresses many of my concerns and provides substantially more evidence, including additional analyses and ablations. In particular, several of these results were already present in the appendix, and the clarification here makes the method better supported overall. I appreciate the additional empirical validation and the effort to clarify the design choices.
> >
> > That said, I find the paper somewhat unfortunate in terms of organization. A large portion of the main text is devoted to PEFT results, which, in my view, are less central, while many important analyses and ablation studies are deferred to the appendix. This presentation choice weakens the clarity of the core contributions and is somewhat regrettable.
> >
> > My remaining concern is primarily about Q2. I still feel that the inclusion of ALS is not well aligned with the core theme of difficulty-aware learning. It appears to be a general SFT trick rather than an integral component of the proposed framework, and the empirical gains seem relatively modest. Its inclusion makes the method feel less clean and somewhat distracts from the main idea.
> >
> > Overall, I feel the core issue of the paper is that it tries to incorporate too many design components at once (e.g., ALS), which makes the method less streamlined and somewhat dilutes the main idea. This also leads to increased complexity and a larger number of hyperparameters, which may limit generality. Moreover, some ablation results (e.g., Fig. 10) show relatively small effects, which also raises questions about whether these additional design components are truly necessary. Despite these concerns, the core idea remains interesting and the rebuttal is helpful, so I am inclined to increase my score to 3. However, these design and presentation choices are somewhat disappointing, as **the paper could have been much stronger with a more focused and streamlined approach**.

---

> > > ### Author Response · Authors · 2026-04-02
> > >
> > > We sincerely appreciate your prompt reply, as well as your recognition that our core idea is interesting and that the responses are helpful. Upon careful consideration of your valuable comments, we understand that your remaining concerns focus on two aspects: 1) the inclusion of multiple components, especially the ALS loss, and 2) the placement of the PEFT results. Accordingly, we humbly and respectfully provide further clarifications on these two points and hope to fully address your concerns.
> > >
> > > > **1. Method design.**
> > >   - **Design coherence:** Our approach is **motivated by three key limitations in existing SFT–RL approaches**: **1)** coarse-grained difficulty modeling; **2)** fixed or manually specified policy allocation; and **3)** suboptimal optimization pathways for SFT and RL. To address these issues, we advocate a training paradigm based on fine-grained difficulty estimation, adaptive sample-level allocation of learning strategies, and refined SFT/RL objectives to improve overall performance. Moreover, **the ALS loss is inherently difficulty-aware**: instead of random or fixed perturbations, it applies adversarial perturbations that maximize sample difficulty, enabling dynamic adaptation to heterogeneous samples and improving decision robustness.
> > >   - **Complexity: Our design takes efficiency into account**: the allocator is a shallow MLP (<0.04M), feature extraction reuses intermediate forward-pass outputs, and both ALS and anti-curriculum weighting (ACW) involve only lightweight perturbation and weighting operations introducing negligible overhead. Moreover, **the method is modular in design, enabling flexible adaptation**. As shown in the further ablations on Qwen2.5-Math-1.5B (table below), **even without ALS, DALSA still outperforms the strongest baseline**. Therefore, in scenarios favoring the simplest approach, one can revert to CE loss while retaining strong performance. Finally, as discussed in our responses to Q4 and Q7, **DALSA achieves lower computational complexity (reduces 12.5%) than existing baselines**.
> > > ||AIME-24|AIME-25|AMC|MATH-500|Olympiad|MMLU-Pro|Overall|
> > > |:-|:-:|:-:|:-:|:-:|:-:|:-:|:-:|
> > > |DALSA|15.1|13.2|48.8|77.6|41.2|33.2|**38.2**|
> > > |w/o ALS|14.9|12.1|48.2|77.1|40.6|32.6|37.6|
> > > |w/o ACW|14.8|12.3|48.4|77.2|40.8|32.7|37.7|
> > > |ReLIFT|14.3|10.0|47.2|76.4|39.6|31.7|36.5|
> > >   - **Hyperparameters:** Our method introduces four hyperparameters—$\lambda_{\rm max}$ and $k$ for ACW, and $\vartheta$ and $\tau$ for ALS—with **no hyperparameters introduced for difficulty modeling or strategy allocation**. As you rightly noted, ACW and ALS can be viewed as general enhancements to RL and SFT; accordingly, **their hyperparameters are highly transferable**. **Fig. 9** shows average performance across tasks, demonstrating the framework’s robustness to hyperparameters. **In practice, the recommended defaults suffice to achieve effective and stable performance across tasks**.
> > >   - **Components’ effectiveness:** The effectiveness of each component in our approach is validated in Tables 9 and 10 of the manuscript. Further ablations assessing the MLP learned routing and the max–min alternating optimization are provided in our response to Q1 of Reviewer 9R91. Moreover, Fig. 10 presents ablations on extracted training features; the following provides results except those in our response to Q4. As you noted, removing any single feature causes only a modest performance drop, indicating that **the allocator does not rely on any single signal but instead effectively integrates multiple complementary features for making decisions**.
> > > ||AIME-25|MATH-500|Olympiad|MMLU-Pro|
> > > |:-|:-:|:-:|:-:|:-:|
> > > |DALSA|16.3|84.1|47.0|58.3|
> > > |w/o reward|16.0|83.7|46.6|58.1|
> > > |w/o perplexity|16.0|83.8|46.7|58.2|
> > > |w/o margin|15.9|83.9|46.8|58.0|
> > > |w/o entropy|15.8|83.7|46.9|57.8|
> > > |w/o logits|16.1|83.9|46.7|57.9|
> > > |w/o length|16.0|84.0|46.8|58.0|
> > >
> > > > **2. Paper organization.**
> > >
> > > We evaluate DALSA under both full fine-tuning and PEFT, **demonstrating its consistent effectiveness**. Following your valuable suggestion that PEFT content is overly extensive in the main text, we have streamlined this section, retaining only the first six rows of Table 3 with the description condensed as: “*DALSA is further evaluated under PEFT across commonsense reasoning, mathematical reasoning, and code generation tasks. Table 3 reports results on eight commonsense reasoning tasks using LLaMA3-8B, with additional results provided in Appendix G. DALSA consistently achieves the best performance…demonstrating its effectiveness in PEFT settings.*” The freed space is then reallocated to other substantive contents, including complexity analysis (Appendix I and corresponding responses) and additional ablations (Appendix L). **Since these materials are already prepared, the reorganization can be carried out efficiently**.
> > >
> > > Once again, we sincerely thank you for your valuable and insightful comments, and we hope that the above responses can adequately address your concerns.

---

### Official Review · Reviewer_tfEv · 2026-03-10

**Soundness:** 3
**Presentation:** 3
**Significance:** 2
**Originality:** 2
**Overall Recommendation:** 4
**Confidence:** 3

**Summary:**

This paper proposes a LLM training framework named DALSA. Its core innovation lies in dynamically assigning the most suitable training strategy to each sample based on its learning difficulty — whether it be SFT, RL, or directly discarding it. Experimental results demonstrate that DALSA outperforms existing methods, including mathematical reasoning and commonsense reasoning, under both full FT and PEFT.

**Compliance With Llm Reviewing Policy:**

Affirmed.

**Final Justification:**

The authors' response and supplementary experiments have addressed most of my concerns; therefore, I have decided to maintain a positive evaluation.

**Key Questions For Authors:**

see weaknesses

**Limitations:**

yes

**Strengths And Weaknesses:**

### Strengths:

1. DALSA employs a learnable allocator to dynamically select the most suitable learning approach based on the difficulty features of each sample, which is more fine-grained and rational than traditional heuristic or alternating strategies.

2. DALSA introduces inverse curriculum weighting for RL and an adversarial label smoothing loss for SFT, and provides theoretical analysis to support their effectiveness.

3. DALSA has been extensively validated across various tasks, including mathematical reasoning, commonsense reasoning, and code generation, under both full FT and PEFT settings.

### Weaknesses:

1. The motivation of DALSA requires further explanation. Specifically, DALSA posits that difficult samples should be trained with SFT, easy samples should be discarded directly, and moderate-difficulty samples should be trained with RL. Do the authors have experimental evidence to support this choice? For instance, are there results from alternative training strategies (difficult sample train with RL and moderate train with SFT)? What is the approximate proportion of these sample categories in each dataset? I would like to know whether there could be cases where $s^{SFT}$, $s^{RL}$, and $s^D$ are equal. Could some case studies also be provided?

2. The core principle (line 60) of DALSA is stated without supporting citations, so we may regard it as a claim. However, I believe the authors should provide comparative experiments, such as applying RL to difficult samples and SFT to moderate difficulty samples, to substantiate this principle or conclusion.

3. The effectiveness of the strategy allocator is highly dependent on whether the six extracted features are effective. If some key features are not captured, or if the feature has noise, it could affect the allocator's decision quality.

4. The core idea of this framework is not entirely novel, as it shares a similar motivation with previous studies like [1].

[1] DAST: Difficulty-Aware Self-Training on Large Language Models

---

> ### Author Rebuttal · Authors · 2026-03-30
>
> Dear Reviewer tfEv,
>
> We sincerely appreciate your thoughtful review and insightful comments. Thank you for recognizing the effectiveness of our learnable allocator, the theoretical soundness of our techniques, and the comprehensive validation of our approach. Below are our detailed responses to your concerns.
> >Q1: Motivation of DALSA.
>
> We humbly address your concerns from the following aspects:
> - Prior work [1,2] validated that SFT excels on samples beyond model knowledge, whereas RL better refines responses within its competence. This motivates our use of difficulty-aware strategies. Following your valuable suggestion, we conduct additional experiments using ReLIFT-style difficulty thresholds: zero-accuracy samples as hard, accuracy >0.9 as easy, and the rest as medium. Model performance is then evaluated under three settings. **Setting I**: SFT for hard, RL for medium, and easy samples discarded. **Setting II**: RL for hard, SFT for medium, easy samples discarded. **Setting III**: SFT for hard, RL for medium and easy. **Setting I achieves the highest performance, validating our motivation**.
> |Setting|AIME-24|AIME-25|AMC|MATH-500|Olympiad|MMLU-Pro|Overall|
> |:-|:-:|:-:|:-:|:-:|:-:|:-:|:-:|
> |I|**28.8**|**24.2**|**65.9**|**88.2**|**58.8**|**54.5**|**53.4**|
> |II|26.1|22.9|62.6|85.8|54.1|52.0|50.6|
> |III|28.3|24.1|65.3|87.8|58.6|54.1|53.0|
> - **Fig. 2(d) shows the evolving proportions of three sample types**: SFT gradually decreases, while RL and DISCARD increase as model improves. This trend highlights DALSA’s ability to adapt strategy allocation to the model’s learning dynamics.
> - As noted in **Footnote 3, we employ a temperature-controlled $\psi$-Softmax to encourage more deterministic allocations**. Even minimal probability differences can be amplified, and thus the rare scenario was not observed in our experiments. Should it arise, our framework applies both SFT and RL training to such samples, ensuring correct system operation.
> - **Tables 12–14 present cases for the three sample types**, with their normalized average probability distributions as follows:
> |SFT|RL|Discard|
> |:-|:-:|:-:|
> |[0.94,0.05,0.01]|[0.05,0.92,0.03]|[0.02,0.06,0.92]|
>
> [1] Learning what reinforcement learning can't: Interleaved online fine-tuning for hardest questions
>
> [2] Metis-rise: RL incentivizes and SFT enhances multimodal reasoning model learning
>
> >Q2: Experiments to substantiate our principle.
>
> Our approach builds on prior work showing that RL excels at refining in-knowledge responses, while SFT is more effective for out-of-knowledge samples [1,2]. We completely agree that experiments are needed to support this claim. Accordingly, we have added those detailed in **our response to Q1 (the first point)** to **Appendix A Preliminary Experiments**, and have added “*This principle is validated by our preliminary experiments in Appendix A.*” to **Section I**.
>
> >Q3: Effectiveness of extracted features.
>
> We analyze the efficacy of the extracted features for strategy allocation from the following aspects:
> - The six training characteristics **capture sample difficulty from complementary perspectives**, spanning intrinsic properties and external reward signals. This multi-dimensional design enhances robustness, as noise in any single feature can be mitigated by others.
> - The allocator is implemented as a learnable MLP rather than a fixed heuristic, functioning **not only as a mapping module but also as a feature selector**. Through alternating optimization, it can prioritize informative features while down-weighting noisy or less relevant signals.
> - Ablation studies in **Appendix M, Fig. 10**, with partial Qwen-7B results below, show that removing individual features causes only gradual performance declines, demonstrating both their contributions and the robustness of the feature set.
> |Data|AIME-25|MATH-500|Olympiad|MMLU-Pro|
> |:-|:-:|:-:|:-:|:-:|
> |DALSA|**16.3**|**84.1**|**47.0**|**58.3**|
> |w/o reward|16.0|83.7|46.6|58.1|
> |w/o perplexity|16.0|83.8|46.7|58.2|
> |w/o margin|15.9|83.9|46.8|58.0|
> |w/o entropy|15.8|83.7|46.9|57.8|
> |w/o logits|16.1|83.9|46.7|57.9|
> |w/o length|16.0|84.0|46.8|58.0|
>
> >Q4: Difference from DAST.
>
> While both our work and DAST operate under a difficulty-aware paradigm, they differ fundamentally in three aspects:
> - **Paradigm**: DAST performs self-training within a single paradigm (e.g., SFT/DPO), whereas we introduce a strategy allocation framework that dynamically assigns samples to three pathways: SFT, RL, or DISCARD.
> - **Difficulty estimation**: DAST uses accuracy as a single heuristic, similar to ReLIFT, whereas DALSA combines six indicators with a learnable allocator for automated difficulty assessment.
> - **Optimization mechanism**: DAST optimizes models on augmented data, whereas we use max–min alternating optimization, with the allocator choosing the beneficial strategy for model updates.
>
> Based on your valuable feedback, we have included DAST which also captures difficulty via accuracy distribution in **Section 2**.

---

> > ### Author Rebuttal · Reviewer_tfEv · 2026-04-04
> >
> > The authors' response and supplementary experiments have addressed most of my concerns; therefore, I have decided to maintain a positive evaluation.

---

> > > ### Author Response · Authors · 2026-04-04
> > >
> > > We sincerely thank you for your prompt reply and for acknowledging that our responses addressed your concerns. We also greatly appreciate your positive recommendation to accept our paper. All your thoughtful and constructive comments and suggestions have played a significant role in improving the quality of our work. Once again, we thank you for your invaluable contributions.

---

### Official Review · Reviewer_9R91 · 2026-03-13

**Soundness:** 3
**Presentation:** 4
**Significance:** 3
**Originality:** 3
**Overall Recommendation:** 5
**Confidence:** 3

**Summary:**

This paper proposes DALSA to adaptively assign different learning strategies, SFT, RL, or discarding, to individual training samples based on their inherent learning difficulty during LLM alignment. DALSA utilizes a learnable strategy allocator guided by multi-faceted training dynamics like perplexity, margin, and entropy. An anti-curriculum weighting scheme for RL and an adversarial label smoothing (ALS) loss for SFT are introduced as complementary regularization.

**Compliance With Llm Reviewing Policy:**

Affirmed.

**Final Justification:**

The rebuttal successfully addressed my concerns by providing crucial ablations and training cost comparison that validate the dynamic allocator's empirical soundness and efficiency. With the experimental design now thoroughly justified alongside the paper's originality, I am raising my score to recommend acceptance.

**Key Questions For Authors:**

See weaknesses.

**Limitations:**

No. The evaluation heavily focuses on objective tasks with clear right/wrong answers (mathematics, code generation, multiple-choice commonsense). It is unclear how well the DALSA framework, particularly its reward mechanism and ALS loss, would generalize to more subjective alignment tasks like helpfulness, harmlessness, or general conversational RLHF.

**Strengths And Weaknesses:**

### Strengths

1. The concept of dynamically routing samples to SFT, RL, or a discard pile via a learnable strategy allocator that co-evolves with the LLM is novel.

2. The paper provides rigorous theoretical analyses demonstrating that the proposed anti-curriculum weighting strategy accelerates convergence compared to standard RL, and the mathematical justification for the ALS loss.

3. The experimental setup is comprehensive, covering models ranging from 1.5B to 70B parameters across both full fine-tuning and PEFT settings.

### Weaknesses

1. The paper identifies the dynamically optimized MLP allocator as its primary contribution. However, the experimental design suffers from severe confounding variables, leaving the efficacy of the Max-Min alternating optimization unproven.
   - While DALSA outperforms ReLIFT (Table 1), DALSA also introduces DISCARD, ALS loss, and anti-curriculum weighting. There is no ablation replacing the MLP with a simple heuristic (e.g., ReLIFT's rules) within the exact same DALSA framework. Without this, it is impossible to know if the performance gain stems from the MLP's learned routing, or simply from the independent effectiveness of ALS, anti-curriculum, and dropping easy data.
   - The authors propose a computationally complex Max-Min alternating optimization (Eq. 6 & 7). Yet, there is no ablation isolating the necessity of this dynamic co-evolution. The paper lacks two baselines where the allocator's weights are lightly pre-trained but frozen, and where the dataset is statically partitioned (offline) using the initial model state. If a static or frozen allocator achieves comparable results, the performance gains may come simply from mixing the six extracted features (margin, perplexity, entropy, etc.) into a routing decision.

2. In Appendix C.2, the authors mention that the allocator is updated asynchronously every 5 LLM updates. An ablation varying this parameter (k=1,5,10,50) is necessary to understand how tightly coupled the allocator's policy must be to the LLM's shifting latent state, and how this impacts training stability.

3. The authors claim DALSA is computationally efficient, citing fewer total GPU hours than baselines like ReLIFT (Table 7). However, this comparison is unfair given DALSA's discard mechanism, skipping the computationally expensive backward pass for up to 30% of the training samples (Fig. 8b). The reported reduction in total training time masks the overhead required to extract the six necessary training characteristics. To transparently assess the true computational cost of the DALSA framework, the authors must provide a detailed wall-clock profiling breakdown that isolates the time spent on LLM generation, feature extraction, allocator updates, and LLM updates.

---

> ### Author Rebuttal · Authors · 2026-03-30
>
> Dear Reviewer 9R91,
>
> We sincerely appreciate your thorough review and insightful comments. We are truly grateful for your recognition of the novelty of our method, the rigor of our theoretical analyses, and the comprehensiveness of our experiments. Below, we provide detailed responses to your concerns.
> >Q1: Confounding aspects of experimental design.
>
> We humbly respond to your concerns from the following aspects:
> - We fully agree that **disentangling the allocator’s contribution from others** is essential. Accordingly, we replace the allocator with a heuristic ReLIFT-style policy: zero accuracy samples are assigned to SFT, those with accuracy > 0.9 are discarded, and the remainder are assigned to RL. The results below show that **the allocator plays a central role in overall performance**. Moreover, ablations in **Table 9** confirm that the other components are also crucial to DALSA’s effectiveness.
> ||AIME-24|AIME-25|AMC|MATH-500|Olympiad|MMLU-Pro|Overall|
> |:-|:-:|:-:|:-:|:-:|:-:|:-:|:-:|
> |ReLIFT|28.3|22.9|65.1|87.9|57.3|53.9|52.6|
> |DALSA (Heuristic)|28.8|24.2|65.9|88.2|58.8|54.5|53.4|
> |DALSA|**29.4**|**26.0**|**66.6**|**88.5**|**59.8**|**55.5**|**54.3**|
> - **The core motivation for alternating optimization is to establish a dynamic feedback loop between model training and strategy allocation**, where the allocator is updated to reflect the evolving optimization landscape and prioritize the most beneficial strategies. In contrast, static partitioning or a frozen allocator assumes constant difficulty and cannot accommodate progressive dynamics. To validate this, we conduct two ablations: **1)** Frozen allocator: Pretrained for 200 steps and then fixed. **2)** Static partitioning: Each sample is assigned (offline) a fixed strategy based on initial LLM states using a trained allocator. **Our max–min optimization achieves the best performance**, highlighting the importance of dynamic allocation. These results have been added to **Appendix L** of our manuscript.
> ||AIME-24|AIME-25|AMC|MATH-500|Olympiad|MMLU-Pro|Overall|
> |:-|:-:|:-:|:-:|:-:|:-:|:-:|:-:|
> |Frozen|28.9|25.1|66.0|88.1|59.0|54.7|53.6|
> |Static|28.5|24.0|65.6|88.2|58.1|54.2|53.1|
> |DALSA|**29.4**|**26.0**|**66.6**|**88.5**|**59.8**|**55.5**|**54.3**|
>
> >Q2: Ablation on update interval.
>
> Following your valuable suggestion, we conduct ablations with varying update intervals, reporting both performance and the average entropy of strategy allocations.
> |Interval|BoolQ|PIQA|SIQA|Hella.|Wino.|ARC-e|ARC-c|OBQA|Overall|Entropy|
> |:-|:-:|:-:|:-:|:-:|:-:|:-:|:-:|:-:|:-:|:-:|
> |1|73.2|86.8|82.1|92.0|85.5|87.0|75.9|82.0|83.1|0.62|
> |5|73.8|87.7|82.2|92.9|85.8|87.1|76.5|82.8|**83.6**|**0.42**|
> |8|73.7|87.5|82.4|92.8|85.8|87.3|76.2|82.9|**83.6**|0.44|
> |10|73.5|87.7|82.2|92.4|85.7|87.1|76.3|82.5|83.4|0.48|
> |50|72.4|86.4|81.8|91.9|85.2|86.8|75.3|81.6|82.7|0.70|
>
> **An interval of 5 balances timely updates with stability**, while very large intervals degrade performance due to outdated allocations.
> >Q3: Wall-clock profiling breakdown.
>
> We respectfully address your concerns as follows:
> - We **profile the average per-step runtime of DALSA and the strongest baseline, ReLIFT**, normalizing our additional times relative to ReLIFT’s total time.
> |Method|LLM forward|Feature extraction|Allocator forward|LLM update|Allocator update|
> |:-|:-:|:-:|:-:|:-:|:-:|
> |ReLIFT|100%|-|-|100%|-|
> |DALSA|88%|3%|<0.1%|81%|<0.1%|
>   - Characteristic computation uses fewer forward passes than RL rollouts, enabling reuse for RL-selected samples. For samples with other strategies, it remains cheaper than ReLIFT, which performs full RL rollouts per sample. Consequently, SFT and DISCARD assignments **reduce LLM forward and update costs to 88% and 81%**, respectively.
>   - Feature extraction and allocator computation contribute **only ~3%** of ReLIFT’s total time, as the former involves simple vector operations and the allocator contains fewer than 0.04M parameters.
> - We further train ReLIFT by discarding the easiest samples at a rate matching DALSA’s DISCARD ratio (5–18% during training, **Fig. 2(d)**), ensuring a comparable number of updated samples. Discarding samples in ReLIFT lowers computation but sacrifices performance, whereas **DALSA achieves better performance-efficacy tradeoff**.
> |Method|GPU Hours|Performance|
> |:-|:-:|:-:|
> |ReLIFT|52×8|52.6|
> |ReLIFT (DISCARD)|47×8|50.9|
> |DALSA|45.5×8|**54.3**|
>
> In summary, sample discarding in DALSA is a difficulty-aware strategy guided by model states, rather than a mere computational shortcut, thereby enabling more efficient and effective learning.
> >Q4: Subjective tasks.
>
> We utilize a binary reward for objective tasks, which can be replaced by a learned reward model for subjective alignment. Similarly, the ALS loss serves as a general SFT regularizer to reduce overfitting. While we focus on objective tasks for reproducibility, consistent with prior SFT-RL studies, we completely agree that extending DALSA to subjective tasks is a promising direction for future work.

---

> > ### Author Rebuttal · Reviewer_9R91 · 2026-04-04
> >
> > Thanks for the detailed rebuttal and new experiments. My concerns have been mostly resolved, so I have raised my score accordingly.

---

> > > ### Author Response · Authors · 2026-04-04
> > >
> > > We sincerely thank you for your prompt reply and for acknowledging that our responses addressed your concerns. We greatly appreciate your decision to raise the score, which has been highly encouraging for our research. Your thoughtful review and constructive feedback have been valuable in significantly improving the quality of our work. Thank you once again for your invaluable contributions.

---

### Official Review · Reviewer_Zd59 · 2026-03-14

**Soundness:** 3
**Presentation:** 3
**Significance:** 3
**Originality:** 3
**Overall Recommendation:** 5
**Confidence:** 3

**Summary:**

The paper proposes to use a trainable allocater for allocating samples for RL and SFT. The allocator considers six types of information including margin, perplexity, entropy, reward mean, logits, and sequence length. The proposed method tackles the non-differentiability of equation (7) via Strait-Through Estimator. The experiments show that the proposed method achieves the overall best performance compared with baselines.

**Compliance With Llm Reviewing Policy:**

Affirmed.

**Final Justification:**

My questions are answered. I regard ALS as an existing engineering trick which is not related to the core idea, but the core idea is still working based on the author's rebuttal https://openreview.net/forum?id=6EX4e6kSd0&noteId=v4hjaA10Qt

**Key Questions For Authors:**

As in weakness.

**Strengths And Weaknesses:**

Strength:

1. The research direction is important: combining RL and SFT is a very important question for LLM finetuning. The proposed method is interesting and reasonable.
2. The author clearly states the difference between exsiting methods and the proposed DALSA: existing methods rely on predefined schedules or heuristic allocation.
3. Experiments show very positive results, and covering varied datasets, baselines, and LLMs.

Weakness:

1. The detail of how the author deal with non-differentiability using Straight-Through Estimator is unclear. The author should give more formulations or details on the implementation. When the parameters of LLM are fixed, how are the parameters of allocater updated?
2. How does the author select the six characteristics is unclear. Does the author try different combinations on the selections?
3. Typos: formulation issue in line 141 when inserting K=1 to the double summation.

---

> ### Author Rebuttal · Authors · 2026-03-30
>
> Dear Reviewer Zd59,
>
> We sincerely thank you for your thoughtful review and constructive comments. We greatly appreciate your recognition of the significance of our research direction, the novelty of our method, and the comprehensiveness and promising nature of our experimental results. Below are our detailed responses to your concerns.
> > Q1: Details on handling non-differentiability via the Straight-Through Estimator.
>
> We fully agree that providing additional explanations of the application of Straight-Through Estimator (STE) within our framework is important.
>
> In the forward pass, the allocator produces a probability distribution $\boldsymbol{\mathcal{S}} = [s^{\rm{SFT}}, s^{\rm{RL}}, s^{\rm{D}}]$ based on the sample features $\boldsymbol{\xi}$. A discrete decision is then made via the $\arg\max$ operation: $$\alpha = \mathbb{I} {\lbrace \arg\max \_ {k \in \lbrace \rm{SFT},\rm{RL},\rm{D}\rbrace } s ^ {k}=\rm{SFT}\rbrace}, \beta=\mathbb{I} \lbrace \arg\max\_{k \in \lbrace\rm{SFT},\rm{RL},\rm{D}\rbrace} s^{k}=\rm{RL}\rbrace.$$
> To enable the propagation of gradients, we introduce the following operations in implementation: $$\alpha = \left(\mathbb{I}\lbrace\arg\max_{k \in \lbrace \rm{SFT, RL, D}\rbrace } s^{k} = \rm{SFT}\rbrace - s^{\rm{SFT}}\right).\text{detach}() + s^{\rm{SFT}}, \beta = \left(\mathbb{I}\lbrace \arg\max_{k \in \lbrace \rm{SFT, RL, D}\rbrace } s^{k} = \rm{RL}\rbrace - s^{\rm{RL}}\right).\text{detach}() + s^{\rm{RL}}.$$
> This ensures that during the forward pass, the $\alpha$ and $\beta$ values are exactly discrete.
>
> During the backward pass, STE treats the discrete $\arg\max$ operation as an identity function, i.e., $$\frac{\partial \alpha}{\partial s^{\rm SFT}} \approx 1, \quad \frac{\partial \beta}{\partial s^{\rm RL}} \approx 1.$$
> Accordingly, the gradient with respect to the allocator parameters $\boldsymbol{\Omega}$ is computed as: $$\nabla\_{\boldsymbol{\Omega}} \mathcal{J}=\mathbb{E}\_{\boldsymbol{x} \sim \mathcal{B}}\left[\frac{\partial \mathcal{J}}{\partial \alpha\_{\boldsymbol{x}}} \cdot \frac{\partial \alpha\_{\boldsymbol{x}}}{\partial s\_{\boldsymbol{x}}^{\rm SFT}} \cdot \frac{\partial s\_{\boldsymbol{x}}^{\rm SFT}}{\partial \boldsymbol{\Omega}}+\frac{\partial \mathcal{J}}{\partial \beta\_{\boldsymbol{x}}} \cdot \frac{\partial \beta\_{\boldsymbol{x}}}{\partial s\_{\boldsymbol{x}}^{\rm RL}} \cdot \frac{\partial s\_{\boldsymbol{x}}^{\rm RL}}{\partial \boldsymbol{\Omega}}\right].$$
> With the STE approximation, this expression simplifies to $$\nabla\_{\boldsymbol{\Omega}} \mathcal{J} \approx \mathbb{E}_{\boldsymbol{x} \sim \mathcal{B}}\left[\mathcal{J}\_{\boldsymbol{x}}^{\mathrm{SFT}} \cdot \nabla\_{\boldsymbol{\Omega}} s\_{\boldsymbol{x}}^{\mathrm{SFT}}+\mathcal{J}\_{\boldsymbol{x}}^{\mathrm{RL}} \cdot \nabla\_{\boldsymbol{\Omega}} s\_{\boldsymbol{x}}^{\mathrm{RL}}\right].$$ In this way, gradients can be smoothly propagated back to the allocator.
>
> These contents have been added to **Appendix C Application of Straight-Through Estimator in DALSA** of our manuscript.
> > Q2: Selection of the six characteristics.
>
> The six characteristics are selected based on established literature on training dynamics and sample difficulty. As discussed in **Section 2**, metrics such as perplexity, accuracy distribution, and length are widely used to assess difficulty in LLMs, while margin, entropy, and logits are commonly adopted in weighting and perturbation studies [1–3]. By integrating these metrics and evaluating their adaptability to LLMs, we identify six features that **provide a multi-dimensional characterization of samples’ learning states**.
>
> To validate the effectiveness of these characteristics, we have conducted ablation studies (**Appendix M**) by evaluating different feature combinations. Specifically, we remove each feature in turn and analyze the resulting performance changes. As shown in **Fig. 10** (partial Qwen-7B results presented below), **each feature contributes to difficulty modeling and strategy assignment**.
> |Data|AIME-24|AIME-25|AMC|MATH-500|
> |:-|:-:|:-:|:-:|:-:|
> |DALSA|**20.6**|**16.3**|**53.4**|**84.1**|
> |w/o reward|20.1|16.0|52.9|83.7|
> |w/o perplexity|20.4|16.0|53.1|83.8|
> |w/o margin|20.3|15.9|53.0|83.9|
> |w/o entropy|20.3|15.8|53.1|83.7|
> |w/o logits|20.5|16.1|52.8|83.9|
> |w/o length|20.0|16.0|53.0|84.0|
>
> Moreover, our allocator has the ability to automatically select and combine features during optimization, making it more robust than manual approaches.
>
> [1] Geometry-aware instance-reweighted adversarial training
>
> [2] Learning imbalanced datasets with label-distribution-aware margin loss
>
> [3] Probability distribution and entropy as a measure of uncertainty
> > Q3: Formulation issue.
>
> We apologize for our negligence and have carefully revised the relevant content as follows: “*the Top-$K$ margin is $\frac{2}{K(K-1)} \sum\_{i=1}^{K-1} \sum\_{j=i+1}^{K} [{p}^{i}(y\_{t}|\boldsymbol{x}, \boldsymbol{y}\_{<t}) - {p}^{j}(y\_{t}|\boldsymbol{x}, \boldsymbol{y}\_{<t})]$, where $K>1$.*”

---

> > ### Author Rebuttal · Reviewer_Zd59 · 2026-04-07
> >
> > My questions are answered. Score raised 1. For ALS, I do regard it as an existing method for increasing the performance, but not related to the core idea. However, the core idea still work based on https://openreview.net/forum?id=6EX4e6kSd0&noteId=VmdJ87wOCS

---

> > > ### Author Response · Authors · 2026-04-08
> > >
> > > We sincerely thank you for your positive reply and for acknowledging that our responses addressed your concerns. We greatly appreciate your decision to raise the score, which has been highly encouraging for our research.
> > >
> > > The core idea of our approach is to assign different learning strategies (SFT, RL, DISCARD) to samples with varying levels of difficulty, enabling a more effective and adaptive training paradigm beyond predefined schedules or heuristics. In our framework, the ALS loss is proposed to further enhance the learning effectiveness along the SFT pathway, mitigate overfitting, and address the issue of label noise.
> > >
> > > We are truly grateful for your recognition and appreciation of the effectiveness of our idea based on both the manuscript and responses. Your thoughtful review and constructive feedback have been valuable in significantly improving the quality of our work. Thank you once again for your invaluable contributions.

---

### Decision · Program_Chairs · 2026-04-30

**Decision:**

Accept (regular)

**Comment:**

This paper has a generally positive score distribution (5543, avg 4.25). Reviewers Zd59 and 9R91 are positive on the novelty of the dynamic strategy-allocation framework and its experimental coverage across multiple model scales, and both raised their scores after the rebuttal; reviewer tfEv's concerns about the core principle and overlap with DAST were fully resolved; reviewer JSbB's concerns about over-engineering, the positioning of the adversarial label smoothing (ALS) component, and the main-text organization were partially resolved. After carefully reviewing the paper, the reviews, the rebuttal, and the discussion, the AC considers that the authors have addressed the main confounding concerns through the rebuttal. The AC recommends acceptance on the condition that the authors incorporate the rebuttal content into the camera-ready, including reframing ALS to clearly separate it from the core allocator, refining the presentation around the "adversarial" wording, and explicitly including the promised additional experiments, results, and discussions from the rebuttal.